

# Investigating the effect of El Niño on nitrous oxide distribution in the Eastern Tropical South Pacific

Qixing Ji[1], Mark A. Altabet[2], Hermann W. Bange[1], Michelle I. Graco[3], Xiao Ma[1], Damian L. Arévalo-Martínez[1], and Damian S. Grundle[1,4]

[1]GEOMAR Helmholtz Centre of Ocean Research Kiel, Kiel, 24103, Germany
[2]School for Marine Science & Technology, University of Massachusetts Dartmouth, New Bedford, Massachusetts, USA
[3]Dirección General de Investigaciones Oceanográficas y cambio Climático, Instituto del Mar del Perú (IMARPE), P.O. Box 22, Callao, Perú
[4]Bermuda Institute of Ocean Sciences, St. George's, GE01, Bermuda

*Correspondence to*: Qixing Ji (qji@geomar.de)

**Abstract.** The open ocean is a major source of atmospheric warming and ozone depleting gas nitrous oxide ($N_2O$). Intense sea-to-air fluxes of $N_2O$ occur in major oceanic upwelling regions such as the Eastern Tropical South Pacific Ocean (ETSP). The ETSP is influenced by the El Niño-Southern Oscillation that leads to inter-annual variations of physical, chemical and biological properties. A strong El Niño was developing in this region in October 2015, during which we investigated the $N_2O$ production pathways and, by comparing to previous non-El Niño years, the effects of El Niño on water column $N_2O$ distributions and fluxes. Analysis of $N_2O$ natural abundance isotopomers suggested that both nitrification and partial denitrification (nitrate and nitrite reduction to $N_2O$) were important $N_2O$ production pathways. Higher than normal sea-surface temperatures were associated with a deepening of the oxycline, while the level of sea surface $N_2O$ supersaturation on the continental shelf was nearly an order of magnitude lower than those of non-El Niño years. Therefore, a significant reduction of $N_2O$ efflux in the ETSP occurred during the 2015 El Niño event. At both offshore and coastal stations, the $N_2O$ concentration profiles during El Niño showed moderate $N_2O$ concentration gradients, and peak $N_2O$ concentrations were deeper than during non-El Niño years; this was likely the result of suppressed upwelling retaining $N_2O$ in subsurface waters. The depth-integrated $N_2O$ concentrations during El Niño were nearly twice as high as those measured in non-El Niño years, indicating subsurface $N_2O$ during El Niño could be a reservoir for intense $N_2O$ effluxes when normal upwelling is resumed after El Niño.

## 1 Introduction

The El Niño-Southern Oscillation (ENSO) is a naturally occurring decadal climate cycle that affects the oceanic and atmospheric conditions across the equatorial Pacific (Philander, 1983). A pronounced effect of ENSO in the ocean is the redistribution of heat flux across the tropical and subtropical Pacific. Generally, the ENSO cycle can be divided into three



phases, El Niño, La Niña and neutral. During El Niño / La Niña years, higher / lower sea surface temperature and deepening / shoaling of the thermocline occur in the Eastern tropical South Pacific (ETSP) (Barber and Chavez, 1983). During El Niño years, upwelling is suppressed in the ETSP, leading to a reduction of upward nutrient fluxes to the surface waters and decreased primary production (Chavez et al., 2003; Ñiquen and Bouchon, 2004).

The ETSP is an oceanic region with intense sea-to-air flux of nitrous oxide ($N_2O$), a strong greenhouse gas and a potent ozone depleting agent (Ravishankara et al., 2009). Diverse microbial processes involved in the production and consumption of $N_2O$ occur in the ETSP, a major oceanic oxygen minimum zones (OMZs) having wide range of $O_2$ concentrations spanning sub-nanomolar level at intermediate depths (Revsbech et al., 2009) to atmospheric saturation at the surface. In the presence of oxygen, $N_2O$ is a by-product during the first step of nitrification, i.e. ammonium ($NH_4^+$) oxidation to nitrite

($NO_2^-$) (Anderson, 1964). Under suboxic and anoxic conditions, $N_2O$ is produced via partial denitrification, i.e. $NO_2^-$ reduction and nitrate ($NO_3^-$) reduction (Codispoti and Christensen, 1985). The dominant biological sink of $N_2O$ in the ocean is the last step of denitrification where $N_2O$ is reduced to $N_2$ under anoxic conditions (Babbin et al., 2015). Recent investigations suggest that $N_2O$ uptake by diazotrophs is a possible $N_2O$ sink at the surface waters (Farías et al., 2013; Cornejo et al., 2015). Its environmental significance awaits further exploration.

Recent modelling efforts have highlighted that the ENSO events could prominently change biogeochemical processes related to nitrogen cycling (Carrasco et al., 2017; Mogollón and Calil, 2017; Yang et al., 2017). During El Niño events in the 1980's, oceanic $N_2O$ fluxes decreased (Cline et al., 1987; Butler et al., 1989), which could be related to changes $O_2$ and organic matter availabilities that are critical environmental factors regulating $N_2O$ production (Elkins et al., 1978; Farías et al., 2009; Arévalo-Martínez et al., 2015; Kock et al., 2016). Here we report water column nitrogen biogeochemistry, $N_2O$

distribution and natural abundance isotopes during October 2015 when a strong El Niño event (recurrence interval > 10 years) was developing (Stramma et al., 2016; Santoso et al., 2017). Stable isotope analyses ($^{15}N$ vs. $^{14}N$ and $^{18}O$ vs. $^{16}O$) were used to determine the pathways of $N_2O$ production and consumption as outlined previously (Yamagishi et al., 2007; Grundle et al., 2017). Recent publications of time-series studies focusing on biogeochemical variations in the ETSP (Gutiérrez et al., 2008; Farías et al., 2015; Graco et al., 2017) and the marine $N_2O$ database (Kock and Bange, 2015) allow us to present the

effects of a strong El Niño event on water column hydrography and $N_2O$ distributions.

## 2 Materials and Methods

### 2.1 Field sampling and laboratory measurements

     The progress and the strength of El Niño was quantified by the Ocean Niño Index (ONI, Figure 1), defined as the running 3-month average sea surface temperature anomaly for the Niño 3.4 region in the east-central tropical Pacific (5˚S –

5˚N, 120˚W – 170˚W). The 2015-16 El Niño was an "extreme El Niño event" indicated by ONI ≥ 0.5 °C. The ASTRA-OMZ SO243 cruise on board the R/V *Sonne* took place between the 5th and 22nd October 2015 from Guayaquil, Ecuador to



Antofagasta, Chile (Figure 2a). In October 2015, the El Niño was still developing with ONI = 2.1 °C, comparable to other strong El Niño events in 1972-73, 1982-83, 1997-98 (Stramma et al., 2016).

The sampling stations are categorized into offshore (Figure 2a in red polygon) and coastal (Figure 2a white polygon) according to their respective water depth: The coastal stations are shallower than 250 m whereas the offshore stations are >
3000 m in depth. Water samples were taken from a 24 x 10L bottle CTD-rosette system. At every station, CTD-Niskin bottles collected water samples at approximately $10 - 20$ depths spanning the observed oxygen concentration range. The CTD system was equipped with two independent sets of sensors for temperature, conductivity (salinity) and oxygen measurements. Calibration for temperature, salinity and oxygen measurements were reported previously, with standard deviations of 0.002°C, 0.0011 PSU, and 0.8 μmol $L^{-1}$ [$O_2$], respectively (Stramma et al., 2016). The detection limit of
dissolved oxygen was ~ 3 μmol $L^{-1}$; the ODZ was operationally defined as water parcels with [$O_2$] < 5 μmol $L^{-1}$, and the boundary of oxygenated layer was defined as [$O_2$] = 20 μmol $L^{-1}$. Dissolved $NO_3^-$ and $NO_2^-$ concentrations were measured at sea with an auto-analyzer (QuAAtro, Seal Analytical, Germany). The detection limit for $NO_3^-$ and $NO_2^-$ was 0.1 and 0.02 μmol $L^{-1}$, respectively. For $N_2O$ concentration measurements, triplicate samples were collected in 20 mL glass vials and were crimp-sealed with butyl stoppers and aluminum caps. Immediately following this, a 10 mL helium headspace was created
and 50 μL of saturated mercuric chloride ($HgCl_2$) solution was added. After an equilibration period of at least 2 hours the headspace sample (10 mL) was measured by a gas chromatograph equipped with an electron capture detector (GC/ECD). The GC was calibrated on a daily basis using dilutions of two standard gas mixtures. The detailed GC/ECD setup and calculation of $N_2O$ concentration were reported previously (Walter et al., 2006; Kock et al., 2016).

Samples for natural abundance $N_2O$ isotopes were collected and preserved with 100 μL of saturated $HgCl_2$ in 160 mL
glass serum bottles with butyl stoppers and aluminum seals. Isotopic measurements of $N_2O$ were carried out at the University of Massachusetts Dartmouth following procedures previously reported (Grundle et al., 2017). In brief, dissolved $N_2O$ was extracted by an automated purge-and-trap system and concentrated with liquid nitrogen. Interfering molecules such as $H_2O$ and $CO_2$ were isolated from $N_2O$ to increase measurement precision. A multi-collector isotope ratio mass spectrometer detected $NO^+$ fragment of $N_2O$ (mass 30, 31, for $\delta^{15}N_\alpha$) and intact $N_2O$ molecule (mass 44, 45 and 46, for
$\delta^{15}N_{bulk}$ and $\delta^{18}O$).

**2.2 Data Analysis**

Water column $N_2O$ saturation was quantified by the $N_2O$ excess ($\Delta N_2O$), defined as the concentration difference between measured and equilibrium values:

$$\Delta N_2O = \left[N_2O\right]_{measured} - \left[N_2O\right]_{equilibrium} \tag{1}$$

The $N_2O$ equilibrium concentration was calculated according to Weiss and Price (1980) with in-situ temperature, salinity and the atmospheric $N_2O$ dry mole fraction in the year of 2015, 328 ppb at 1 atmospheric pressure (Blasing, 2016).



The $N_2O$ efflux from the ocean to the atmosphere was calculated as the product of $N_2O$ excess and gas transfer coefficient ($k_w$, cm hr$^{-1}$) that was derived according to empirical relationship proposed by Wanninkhof (2014):

$$k_w = 0.251 \times U_{10}^2 \times \left(Sc/660\right)^{-0.5}$$

(2)

where $U_{10}$ denotes wind speed (m s$^{-1}$) at 10 m above sea surface, Sc denotes the Schmidt number for $N_2O$ under in-situ temperature (Wanninkhof, 2014).

The notations for $N_2O$ isotopomer ratios (δX) are defined as the relative difference between isotopic ratio (R) of sample and reference material:

$$\delta X = \frac{R_{sample}}{R_{reference}} - 1$$

(3)

where X denotes $^{15}N_\alpha$, $^{15}N_\beta$ and $^{18}O$, R denotes the $^{15}N/^{14}N$ at the central (α), terminal (β) and $^{18}O/^{16}O$ at oxygen positions of the $N_2O$ molecule (see supplementary). Furthermore, the $\delta^{15}N_{bulk}$ (conventionally $\delta^{15}N$) and site preference (SP) are defined as follows:

$$\delta^{15}N_{bulk} = \frac{\delta^{15}N_\alpha + \delta^{15}N_\beta}{2}$$

(4)

$$SP = \frac{\delta^{15}N_\alpha - \delta^{15}N_\beta}{2}$$

(5)

The value of δX is expressed as permil (‰) deviation relative to a set of reference materials: atmospheric $N_2$ for $\delta^{15}N_{bulk}$ $\delta^{15}N_\alpha$ and $\delta^{15}N_\beta$ (Mohn et al., 2014), and Vienna standard mean ocean water (VSMOW) for $\delta^{18}O$.

## 2.3 Additional datasets

The twice-weekly, 50-km resolution of sea surface temperature anomaly from NOAA's Satellite Coral Bleaching Monitoring Datasets (https://coralreefwatch.noaa.gov/satellite/ methodology/methodology.php) were used to quantify the sea-surface temperature difference of the ETSP during October 2015 relative to 1985 – 1993. For $N_2O$ flux calculations, instantaneous wind speed data at each of our sampling locations were acquired from shipboard metrological measurements. Seawater $N_2O$ and oxygen concentrations from previous sampling campaigns in the ETSP were extracted from the MEMENTO database (Kock and Bange, 2015). Specifically, data from the following cruises were used for comparison between El Niño and non-El Niño years: NITROP-85 (February 1985), M77/3 (January 2009), Callao Time Series Transect (October 2011), M90 (November 2012), M91 (December 2012), AT26-26 (January 2015). The ONI of these years (1985, 2009, 2011, 2012, Figure 1) indicated that, 1985 and 2011 are considered as weak "La Niña" years, whereas 2009 and 2012 are considered "neutral" years.



## 3 Results

### 3.1 Hydrography, distribution of oxygen and inorganic nitrogen

An extreme El Niño event during 2015-16 impacted the ETSP with a relatively high sea surface temperature anomaly, especially at the equatorial region (2 °S – 2 °N and 80 – 90 °W) where the highest anomaly between 3 and 5 °C was observed
at offshore waters (Figure 2a). The El Niño-induced warming effect decreased southwards. Between 5 °S and 12 °S, the temperature anomaly was 2 – 3 °C. South of 12 °S the anomaly was generally < 1 °C. The shelf areas (7 °S – 14 °S) had a progressively lower temperature anomaly southwards; > 1.5 °C and < 1 °C north and south of 12 °S, respectively.

Five water masses, based on their thermohaline indices (Strub et al., 1998; Silva et al., 2009) were identified (Figure 2b). The northward-flowing Antarctic Intermediate water (AAIW, T = 3 – 5 °C, S ≈ 34.5) was found at depths below 1000
m. The Equatorial subsurface water (ESSW, T = 8 – 12 °C, S = 34.7 – 34.9) was near the Peruvian coast at depths between 300 and 400 m. Above the continental slope (water depth < 250 m), the colder Peru coastal water (PCW, T < 19°C, S ≈ 35) occupied 30 – 250 m, whereas the warmer subtropical surface water (STSW, T > 18.5 °C, S > 34.9) was found at depth < 30 m. The surface water north of the equator consisted of the tropical surface water (TSW), which had high temperature and low salinity (T > 25 °C, S < 33.5) due to excess precipitation.

Along the offshore section, the upper oxycline boundary ($[O_2]$ = 20 µmol $L^{-1}$ isoline) was at 250 – 300 m along the equator at 85.5 °W, and the ODZ ($[O_2]$ < 5 µmol $L^{-1}$) appeared near 10 °S (Figure 3a). The southward shoaling of the oxycline, thickening of the ODZ and shoaling of the isoline with $[NO_3^-]$ = 20 µmol $L^{-1}$ were observed south of 10 °S (Figure 3a and 3b), where the thickness of the ODZ was ~ 300 m. The top of the ODZ reached ~125 m between 13 °S and 16 °S. Significant accumulation of $NO_2^-$ (>1 µmol $L^{-1}$) occurred south of 10 °S between 30 and 400 meters within the ODZ (Figure
3c), corresponding to lower $NO_3^-$ concentrations (Figure 3b). The highest $NO_2^-$ concentration (9.4 µmol $L^{-1}$) was recorded at 200 m at 15.7 °S.

Along the coastal section, the surface (upper 10 m) $O_2$ concentrations were below saturation at all sampling stations (50 – 97 % saturation, calculated according to Garcia and Gordon (1992)). Surface $O_2$ concentrations were 165 – 217 µmol $L^{-1}$ north of 10 °S and gradually decreased to 135 – 190 µmol $L^{-1}$ between 10°S and 12.5°S, and to 120 µmol $L^{-1}$ south of 14 °S
(Figure 3d). The shoaling of the 20 µmol $L^{-1}$ $O_2$ isoline was observed south of 9 °S. The top of the ODZ was found at 200 m, 150 m and 80 m at 11 °S, 12 °S and 14 °S, respectively. The surface $NO_3^-$ concentrations were 11 – 23 µmol $L^{-1}$ between 9 °S and 16°S, and the 20 µmol $L^{-1}$ $NO_3^-$ isoline was at 0 – 20 m (Figure 3e). Water column $NO_2^-$ concentrations at coastal stations were generally below 1 µmol $L^{-1}$, with the exception of the station at 14.0 °S where $NO_2^-$ concentrations reached 1.2 µmol $L^{-1}$ below 200 m (Figure 3f).






### 3.2 Water column N₂O concentrations and isotopes

Along the offshore section, the water column N$_2$O distributions showed a southward increase of surface N$_2$O concentrations and southward decrease of subsurface N$_2$O maxima (Figure 4a). The equatorial region (1 °N to 2.5 °S, 85.5 °W) had subsurface high N$_2$O concentrations (up to 93 nmol L$^{-1}$) at intermediate depths (200 – 550 m); water column

$\delta^{15}$N, SP and $\delta^{18}$O generally increased with depth (Figure 4b, 4c and 4d); at the subsurface N$_2$O concentration maximum, $\delta^{15}$N, SP and $\delta^{18}$O were ~ 6 ‰, 13 – 17 ‰, and 45 – 50‰, respectively. Two N$_2$O concentration maxima were observed at stations south of 10 °S where the ODZ was formed. Near 10 °S, two N$_2$O concentration maxima (70 ± 6 nmol L$^{-1}$) occurred between 200 and 600 m; and a local concentration minimum (~ 30 nmol L$^{-1}$) occurred within the ODZ at 400 m, associated with high $\delta^{15}$N (8 – 10 ‰), SP (20 – 30‰) and $\delta^{18}$O (60 – 70‰). Near 13°S, a shallow N$_2$O concentration maximum (~ 80

nmol L$^{-1}$) occurred at ~100 m, and a local N$_2$O concentration minimum (18 nmol L$^{-1}$) occurred at 350 m. Between 14 °S and 16 °S, the lowest (< 10 nmol L$^{-1}$) N$_2$O concentrations were observed at 200 – 400 m within the ODZ, where the highest values of $\delta^{15}$N (> 10 ‰), SP (30 – 40‰) and $\delta^{18}$O (> 60 ‰) were observed.

Along the coastal section, a southward increase of surface N$_2$O concentration (20 nmol L$^{-1}$ north of 11 °S and > 40 nmol L$^{-1}$ south of 13 °S) was observed, coinciding with southward shoaling of the ODZ (Figure 4e). Subsurface maximum N$_2$O

concentrations were observed below 200 m near 10.7 °S, and at 80 – 90 m south of 12 °S, where ODZ was formed. The $\delta^{15}$N in coastal waters were between 2.5 and 5 ‰, with lower values at stations south of 14 °S (Figure 4f). SP was lower (< 0 ‰) at the surface (< 10 m) near 9 °S and at 50 – 150 m near 11 °S; higher SP (10 – 20 ‰) was observed south of 14 °S (Figure 4g). The $\delta^{18}$O values were 45 – 60 ‰; higher $\delta^{18}$O (> 55 ‰) were observed within the ODZ below 200 m at 14 °S and below 100 m at 15.3 °S (Figure 4h).

### 3.3 Excess N₂O and N₂O flux to the atmosphere

Both the offshore and coastal stations showed N$_2$O supersaturation in the top 10 m of surface water, and coastal stations had higher $\Delta$N$_2$O concentrations (15 – 50 nmol L$^{-1}$) than those of offshore stations (4 – 8 nmol L$^{-1}$). Subsurface $\Delta$N$_2$O along the offshore section had higher concentrations (70 – 80 nmol L$^{-1}$) at the equatorial regions than $\Delta$N$_2$O

concentrations (40 – 60 nmol L$^{-1}$) at stations located south of 10 °S (Figure 5a). Near 15 °S, subsurface N$_2$O undersaturation was observed; $\Delta$N$_2$O concentrations were -4 – 0 nmol L$^{-1}$ at intermediate depths (200 – 400 m) within the ODZ ([O$_2$] < 5 µmol L$^{-1}$). Along the coastal section, a southward increase of surface and subsurface (50 – 200 m) $\Delta$N$_2$O was observed (Figure 5b). Subsurface maximum $\Delta$N$_2$O concentrations were > 60 nmol L$^{-1}$, and occurring at the periphery of ODZ (~ 200 m near 10 °S and < 100 m south of 12 °S). Undersaturation of N$_2$O ($\Delta$N$_2$O < 0) did not occur in any coastal stations. The N$_2$O

fluxes from the coastal stations were 23 – 108 µmol m$^{-2}$ d$^{-1}$, nearly two folds of the offshore fluxes (7 – 50 µmol m$^{-2}$ d$^{-1}$, Figure 5c). The highest flux occurred at a coastal station at 14.4 °S, 77.3 °W, coinciding with the highest surface $\Delta$N$_2$O (50 nmol L$^{-1}$).



## 4 Discussion

The ETSP is one of the world's major OMZs having active $N_2O$ production and intense efflux to the atmosphere (Arévalo-Martínez et al., 2015; Kock et al., 2016). The gradient spanning from fully oxygenated conditions to anoxia creates suitable conditions for $N_2O$ production and consumption, which causes the co-existence of water column $N_2O$

supersaturation and undersaturation (Codispoti and Christensen, 1985). To identify the $N_2O$ cycling pathways, we input $N_2O$ isotopomer measurements into a simple mass balance model (section 4.1). Quantitative relationships linking $O_2$, $NO_3^-$ and $N_2O$ were examined to characterize the effect of oxygenation on $N_2O$ production from $NH_4^+$ oxidation (section 4.2). Previously measured $N_2O$ concentrations from the ETSP (MEMENTO database, Kock and Bange (2015)) were compared to data from this study to investigate the difference in water column $N_2O$ distribution and effluxes between El Niño and non-El

Niño years (section 4.3), which would better constrain the natural variability of $N_2O$ cycling in the ETSP.

### 4.1 N₂O cycling pathways inferred from natural abundance isotopic and isotopomeric signatures

The analyses of natural abundance isotopomers quantify the substitutions of nitrogen and oxygen isotopes ($^{15}N$, $^{14}N$, $^{18}O$ and $^{16}O$) occurring on the linear asymmetric $N_2O$ molecule (Yoshida and Toyoda, 2000), and can be used to identify

potential production and consumption pathways (Yamagishi et al., 2007; Grundle et al., 2017). The production of $N_2O$ in an isolated water body follows mass conservation of the respective isotopes. The mass balance model proposed by Fujii et al. (2013) quantified the isotopic signature of $N_2O$ produced within the water mass ($\delta_{produced}$) by the linear regression of the inverse $N_2O$ concentration ($1/[N_2O]_{measured}$) and the respective isotope values ($\delta_{observed}$):

$$\delta_{observed} = \frac{1}{[N_2O]_{measured}} \times (\delta_{initial} - \delta_{produced}) \times [N_2O]_{initial} + \delta_{produced}$$

(6)

where $[N_2O]_{initial}$ and $\delta_{initial}$ refer to source water $N_2O$ concentration and isotopic signature, respectively. It has been shown that SP is indicative of $N_2O$ production pathways, because SP is independent of isotopic values of $N_2O$ production substrates; generally, $N_2O$ produced via $NH_4^+$ oxidation and partial denitrification have distinctive SP values of $30 \pm 5$ ‰ and $0 \pm 5$ ‰, respectively (Toyoda et al., 2011). Thus, $N_2O$ production processes can be qualitatively characterized by means of $SP_{produced}$. We further identified four water bodies (coastal and offshore stations combined) from shallow to deeper depths

with distinctive features such as $O_2$, $NO_2^-$ concentrations and depths (Table 1) to discuss $N_2O$ cycling pathways as follows.

(1) Upper oxycline and surface (Figure S1a): $[O_2] > 20$ μmol L$^{-1}$. All the samples were from $< 200$ m (data not shown), $N_2O$ production from this water body could actively contribute to atmospheric efflux. The samples had variable SP values (-9 – 34 ‰); some coastal samples had the lowest water column SP values ever reported (-9 ‰, Figure 4g). The low $SP_{produced}$ (6.4±1.9) indicates that both nitrification and denitrification were sources of $N_2O$ to the upper oxycline, with the majority

appearing to come from denitrification. Given that the $O_2$ concentrations were too high for denitrification to proceed locally in the upper oxycline and the surface (Codispoti and Christensen, 1985), the SP signature in this water body was a mixture of





local nitrification and upwelling/diffusion from suboxic zones of the ODZ proper. Thus, denitrification and nitrification both contribute to $N_2O$ effluxes in the ETSP-OMZ, consistent with a previous study which focused on the coastal regions between ~12 – 14 ˚S (Bourbonnais et al., 2017).

(2) $N_2O$ peak (Figure S1b): $[O_2] < 20$ μmol $L^{-1}$ and $[NO_2^-] < 1$ μmol $L^{-1}$. Generally the samples were from $N_2O$ concentration maxima near the upper boundary of the ODZ. The $SP_{produced}$ is relatively low (8.3±3.0 ‰) at this suboxic water body ($[O_2] < 20$ μmol $L^{-1}$), which allowed $N_2O$ production from denitrification while inhibited $N_2O$ consumption (Bonin et al., 1989; Farías et al., 2009). With the $SP_{produced}$, we conclude that water column $N_2O$ maximum was mainly attributed to partial denitrification (i.e. $NO_2^-$ and $NO_3^-$ reduction), with minor contribution from nitrification. This is consistent with previous $^{15}N$ tracer incubation experiments demonstrating the coincidence between local $N_2O$ concentration maximum and high rates of $N_2O$ production from $NO_2^-$ and $NO_3^-$ reduction that are 10 – 100 fold higher than the rate of $N_2O$ production from $NH_4^+$ oxidation (Ji et al., 2015).

(3) Oxygen deficient zone (Figure S1c): $[O_2] < 5$ μmol $L^{-1}$ and $[NO_2^-] > 1$ μmol $L^{-1}$. Accumulation of $NO_2^-$ within the anoxic layer is prominent feature of ODZ, where $N_2O$ consumption occurs (Codispoti and Christensen, 1985). The isotopic signature of "produced $N_2O$" had distinctively high $\delta^{15}N_{bulk}$ (8.5 ‰), and $\delta^{18}O$ (71 ‰, Table 1 and Figure S2), and this is indicative of $N_2O$ reduction to $N_2$ which results in an isotope enrichment of the remaining $N_2O$ pool in the process of N-O bond breakage (Toyoda et al., 2017). The SP signature was also high (39.9 ‰). While $NH_4^+$ oxidation can produce $N_2O$ with similar SP values, we rule this out given the low $O_2$ concentrations which were observed (Peng et al., 2016). instead, similar to the high $\delta^{15}N_{bulk}$ and $\delta^{18}O$ values which were observed, we suggest that the high SP values which were recorded in the ODZ, where $N_2O$ undersaturation occurred, were also a result of $N_2O$ consumption, as reduction of $N_2O$ can also result in high SP values (Popp et al., 2002; Well et al., 2005; Mothet et al., 2013). Based on the observed $\delta^{15}N_{bulk}$, $\delta^{18}O$ and SP values of $N_2O$, we conclude that $N_2O$ consumption was the predominant $N_2O$ cycling pathway in the water body with $[O_2] < 5$ μmol $L^{-1}$ and $[NO_2^-] > 1$ μmol $L^{-1}$ in the ETSP.

(4) Intermediate waters (Figure S1d): Samples from depths 500 – 1000 m. Generally, the $N_2O$ concentrations peaks below the oxygen minimum layer at the offshore waters can be found in this water body (Figure 4a). From the linear regression, the $SP_{produced}$ is 15.6±4.1 ‰. The samples had $[O_2] = 5 – 70$ μmol $L^{-1}$, suitable for $N_2O$ production from both nitrification and denitrification. Downward mixing and diffusion from ODZ is unlikely because the ETSP is a major upwelling region and ODZ samples had high SP values (see next paragraph). We conclude that localized $N_2O$ production from nitrification and denitrification are important pathways in this region of the water column, and probably contributed to $N_2O$ concentrations maxima in intermediate waters, as reported by Carrasco et al. (2017).

There are some limitations of the isotopomers-based analysis of potential $N_2O$ production pathways. (1) Constant atmospheric exchange at the surface and mixed layer, and mesoscale eddy activities at intermediate waters (Arévalo-Martínez et al., 2016) could affect the $SP_{produced}$ from localized $N_2O$ production. Nevertheless our conclusion of denitrification being important pathway remains valid. As a comparison, water bodies were divided by potential density (i.e. $\sigma_\theta > 27$ kg $m^{-3}$, 26 – 27 kg $m^{-3}$, 25 – 26 kg $m^{-3}$, < 25 kg $m^{-3}$) and showing $SP_{produced}$ of 5.0 – 11.1 ‰. (2) The rates of $N_2O$



production was not derived due to the lack of complimentary dataset (i.e. nitrate and nitrite isotopes) and thus we are not able to investigate the change of $N_2O$ production rates during nitrification and denitrification that are affected by El Niño-induced lower export production (in comparison to non-El Niño years, Espinoza-Morriberón et al. (2017)).

## 5   4.2 The effect of $O_2$ on $N_2O$ production from $NH_4^+$ oxidation

The oxygenated surface mixed layer is constantly in direct contact with the atmosphere. Thus, $N_2O$ production via $NH_4^+$ oxidation is important to oceanic $N_2O$ fluxes. During $NH_4^+$ oxidation to $NO_2^-$, the effectiveness of $N_2O$ production in oxygenated waters can be quantified with the $N_2O$ yield, which is defined as the molar nitrogen ratio of $N_2O$ produced and $NH_4^+$ oxidized. In oxygenated waters, the near absence of $NH_4^+$ and $NO_2^-$ suggest the amount of $NH_4^+$ oxidized produces

equal amounts of $NO_3^-$ within measurement error. Rees et al. (2011) and Grundle et al. (2012) computed the $N_2O$ yield by deriving the slope of the linear regression of $\Delta N_2O$-$NO_3^-$ relationship. The $\Delta N_2O$ data from all sampling stations during October 2015 showed that $\Delta N_2O$ increases with increasing $NO_3^-$ concentrations and decreasing $O_2$ concentrations (Figure 6). The samples from the upper oxycline ($[O_2] > 20$ µmol $L^{-1}$ and depth < 500 m) showed moderate increase of $\Delta N_2O$ ($0 - 20$ nmol $L^{-1}$) when $[NO_3^-] < 20$ µmol $L^{-1}$. At $[NO_3^-] > 20$ µmol $L^{-1}$, substantial increase of $\Delta N_2O$ ($20 - 75$ nmol $L^{-1}$) was

observed. Here, to avoid sampling the ODZ where suboxic condition stimulates $N_2O$ production from partial denitrification, only data from the upper oxycline were used to perform linear regression. The slope of the regression at $[NO_3^-] < 20$ µmol $L^{-1}$ (corresponding to $[O_2] > 100$ µmol $L^{-1}$) is $0.85 \pm 0.11$, indicating that $0.085 \pm 0.011$ nmol of $N_2O$ is produced for every µmol of $NO_3^-$ produced (or $NH_4^+$ oxidized), equating to a molar nitrogen yield (mol $N_2O$-N produced / mol $NO_3^-$ produced) of $0.17 \pm 0.02$ %. At $[NO_3^-] > 20$ µmol $L^{-1}$ (corresponding to $[O_2] < 100$ µmol $L^{-1}$) the yield increases to $0.85 \pm 0.13$ %.

These $N_2O$ yield estimates are generally comparable to previously reported values ($0.04 - 1.6$ %) in the ETSP (Elkins et al., 1978; Ji et al., 2015), and indicating that potential $N_2O$ production from $NH_4^+$ oxidation decreases with water column oxygenation due to intrusion of oxygen-rich water masses (Llanillo et al., 2013; Graco et al., 2017). As discussed earlier, the oxycline samples were probably influenced by mixing of suboxic water with active denitrification producing high $N_2O$ concentrations and low $NO_3^-$ concentrations; the $N_2O$ yield estimates here are thus spatially and temporally integrated. As a

comparison, [15]N tracer incubation method directly measured 0.04 % $N_2O$ yield during $NH_4^+$ oxidation at $[O_2] > 100$ µmol $L^{-1}$ (Ji et al., 2015).

## 4.3 $N_2O$ distribution and fluxes during El Niño

Excess $N_2O$ ($\Delta N_2O$) in surface waters is one of the principal factors regulating $N_2O$ fluxes. To evaluate the effect of

strong El Niño events on oceanic $N_2O$ fluxes, we compare surface and water column $\Delta N_2O$ concentrations in shelf waters (< 300 m depth) along $8 - 16$ °S during El Niño (October 2015) and "neutral" conditions (December 2012, from Kock et al. (2016)). Both data sets revealed that higher surface $\Delta N_2O$ concentrations and thus higher potential $N_2O$ efflux occurred at



near-shore waters. Generally, the surface $\Delta N_2O$ concentrations in October 2015 (Figure 7a) were lower than those of December 2012 (Figure 7d); highest surface $\Delta N_2O$ concentrations were 50 and 250 nmol $L^{-1}$ in 2015 and 2012, respectively. The region of high surface $\Delta N_2O$ occurred at near ~ 14 °S and ~ 10 °S in 2015 and in 2012, respectively. It appears that $N_2O$ efflux was significantly reduced during El Niño; in October 2015, coastal water had $N_2O$ flux of 23 – 108 μmol $m^{-2}$ $d^{-1}$ (Figure 5c), much lower than that of December 2012 having 459 – 1825 μmol $m^{-2}$ $d^{-1}$ (Arévalo-Martínez et al., 2015).

Suppressed upwelling or increased downwelling during El Niño events, as observed in both observational and model studies (Llanillo et al., 2013; Graco et al., 2017; Mogollón and Calil, 2017), can directly and indirectly affect $N_2O$ fluxes to the atmosphere: First, reduced upward transport of subsurface $N_2O$-rich water not only decreased surface $\Delta N_2O$, but also increased subsurface $\Delta N_2O$, which is illustrated by the comparative observation of higher subsurface $\Delta N_2O$ concentrations in coastal waters in October 2015 (Figure 7b, 7c) than those in December 2012 (Figure 7e, 7f). Second, because the oxygen sensitivity of the denitrification sequence increases with each step (Körner and Zumft, 1989), El Niño-induced water column oxygenation inhibited $N_2O$ consumption within the ODZ (bounded by $[O_2]$ = 5 μmol $L^{-1}$ isoline), as demonstrated by the disappearance of $N_2O$ undersaturation ($\Delta N_2O < 0$) in coastal water in 2015 (Figure 7b, 7c), contrasting to water column $N_2O$ undersaturation occurring at 100 m at 13 – 14 °S in December 2012 (Figure 7e, 7f). Third, as shown in this study, the deepening and expansion of the suboxic zone caused by the El Niño event ($[O_2]$ = 5 – 20 μmol $L^{-1}$) stimulated subsurface $N_2O$ production via denitrification, as demonstrated by the close spatial coupling between local maximum $\Delta N_2O$ concentrations and the oxycline ($[O_2]$ = 5 and 20 μmol $L^{-1}$ isolines, Figure 7b and 7e). Lastly, upwelling of oxygen-rich water along the Peruvian coast, especially north of 12 °S (Stramma et al., 2016), inhibited local $N_2O$ production and caused the southward relocation of surface $\Delta N_2O$ "hot spots".

The decrease of surface $\Delta N_2O$ concentration during El Niño was associated with an increase of subsurface $N_2O$ concentrations. Water column $\Delta N_2O$ concentration profiles at expanded temporal and spatial coverage (see Figure 2a for station coordinates) were compared within the same season between El Niño and non-El Niño years (Figure 8). We included data from January 2015, which had the highest ONI during austral summer than any other years. Generally, subsurface $\Delta N_2O$ concentration peaks were observed at deeper depths during 2015. Offshore stations had higher subsurface peak $\Delta N_2O$ concentrations during El Niño (Figure 8a, 8b), except at station C where the peak concentration during October 2015 was comparable to that of December 2012 (Figure 8c). At coastal stations D and E, higher $\Delta N_2O$ concentrations were found below 50 m but peak $\Delta N_2O$ concentrations were lower during El Niño years (Figure 8d, 8e). In the southernmost coastal station F, the peak $\Delta N_2O$ concentration was higher in 2015 than that of 1985; both were found at similar depths at ~ 60 m. The increase of subsurface $N_2O$ concentrations during El Niño resulted in OMZ water column retaining larger amount of $N_2O$, as shown by higher depth-integrated $N_2O$ concentrations during El Niño years than normal years in both coastal and offshore waters (Figure 9).

In all, the apparent decrease in $N_2O$ efflux during the El Niño event in the tropical Pacific, as shown in this study and others (Cline et al., 1987; Butler et al., 1989) is the result of complex physical and biochemical changes. The above comparative analyses are simple due to limited data availability. Consequently, these following aspects are yet to be





resolved: (1) It is unclear how offshore N$_2$O fluxes vary from "neutral" to El Niño years. Current ΔN$_2$O profiles show higher surface ΔN$_2$O concentrations at station A and B in 2015 (Figure 8a and 8b), whereas the surface ΔN$_2$O was lower in 2015 at station C (Figure 8c). A zonal (east-west) section near 12 °S showed slightly higher offshore surface ΔN$_2$O in 2015 (~ 5 nmol L$^{-1}$, Figure 7c) than in 2012 (~ 1 nmol L$^{-1}$, Figure 7f). The decrease in coastal N$_2$O fluxes during El Niño could be compensated by increase in offshore fluxes. (2) The southward relocation of high surface ΔN$_2$O from neutral to El Niño years (Figure 7a and 7d) possibly results in higher surface ΔN$_2$O hence higher N$_2$O flux in southern region of ETSP (e.g. south of 16 °S, Figure 8f). (3) It is possible that once the normal upwelling is resumed after the El Niño event, N$_2$O produced and retained in the subsurface layer in coastal and offshore waters could be a potential reservoir contributing to high N$_2$O fluxes. (4) The co-occurrence of El Niño and mesoscale eddy formation along the Peruvian coast will have complicated effects on N$_2$O fluxes, which remains unexplored.

## 5 Conclusions

The eastern tropical South Pacific Ocean is a major upwelling region that is ideal to study the effect of strong El Niño events on N$_2$O efflux and associated water column biogeochemistry. During a developing strong El Niño event in October 2015, a more pronounced warming effect occurred at lower latitudes in the ETSP. In comparison to conditions in December 2012 (non-El Niño), the depths of the oxygen deficient zone's upper boundary at lower latitudes were deeper in October 2015, coinciding with lower peak N$_2$O concentrations. Shelf N$_2$O effluxes were significantly lower during 2015 El Niño as a result of lower surface levels of N$_2$O supersaturation. However, a change of upwelling pattern appeared to cause higher subsurface N$_2$O concentrations and doubled the depth-integrated N$_2$O concentrations during El Niño than in other non-El Niño years. Natural abundance isotopic and isotopomer analysis indicated that both nitrification and denitrification are important pathways for N$_2$O production, and denitrification-derived N$_2$O probably contributes to the efflux to the atmosphere. Decreased N$_2$O efflux and subsurface accumulation during strong El Niño events is likely the result of suppressed upwelling and water column oxygenation. However, the complex spatial and temporal patterns of El Niño-induced N$_2$O distribution in ETSP remain to be explored.



# Figures and tables

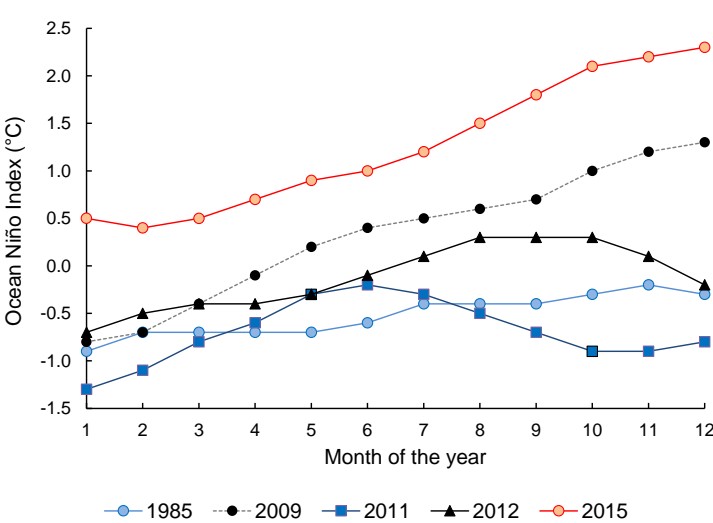

**Figure 1: Ocean Niño Index of year 1985 (weak La Niña), 2009 (neutral), 2011 (weak La Niña), 2012 (neutral) and 2015 (strong El Niño). Data was downloaded from:**
5  **http://origin.cpc.ncep.noaa.gov/products/analysis_monitoring/ensostuff/ONI_v5.php.**

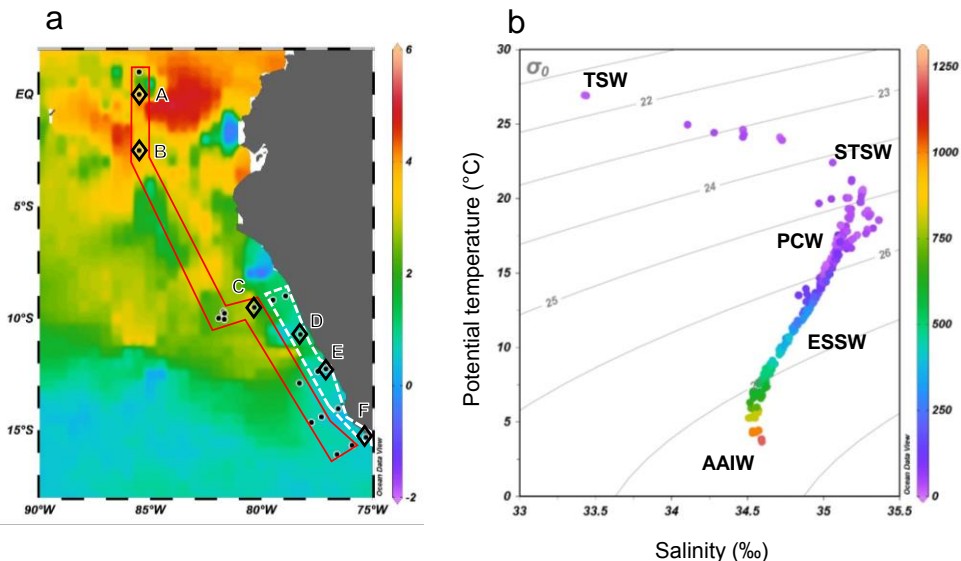

**Figure 2: (a) Monthly mean sea surface temperature anomaly (°C) of October 2015 from NOAA's Satellite Coral Bleaching Monitoring Datasets. Sampling stations (filled circles) are categorized as "offshore" (in red polygon) and "coastal" sections (in white polygon). Comparative analyses of water column $N_2O$ (see section 4.3) were performed at**
10  **stations A – E (open diamonds). (b) Potential temperature – salinity diagram, with corresponding depths (meters, colour bar on right) and potential density ($\sigma_0$, kg m$^{-3}$) of all sampling stations. Five water masses are shown: Tropical surface water (TSW), Subtropical surface water (STSW), Peru coastal water (PCW), Equatorial subsurface water (ESSW) and Antarctic intermediate water (AAIW).**





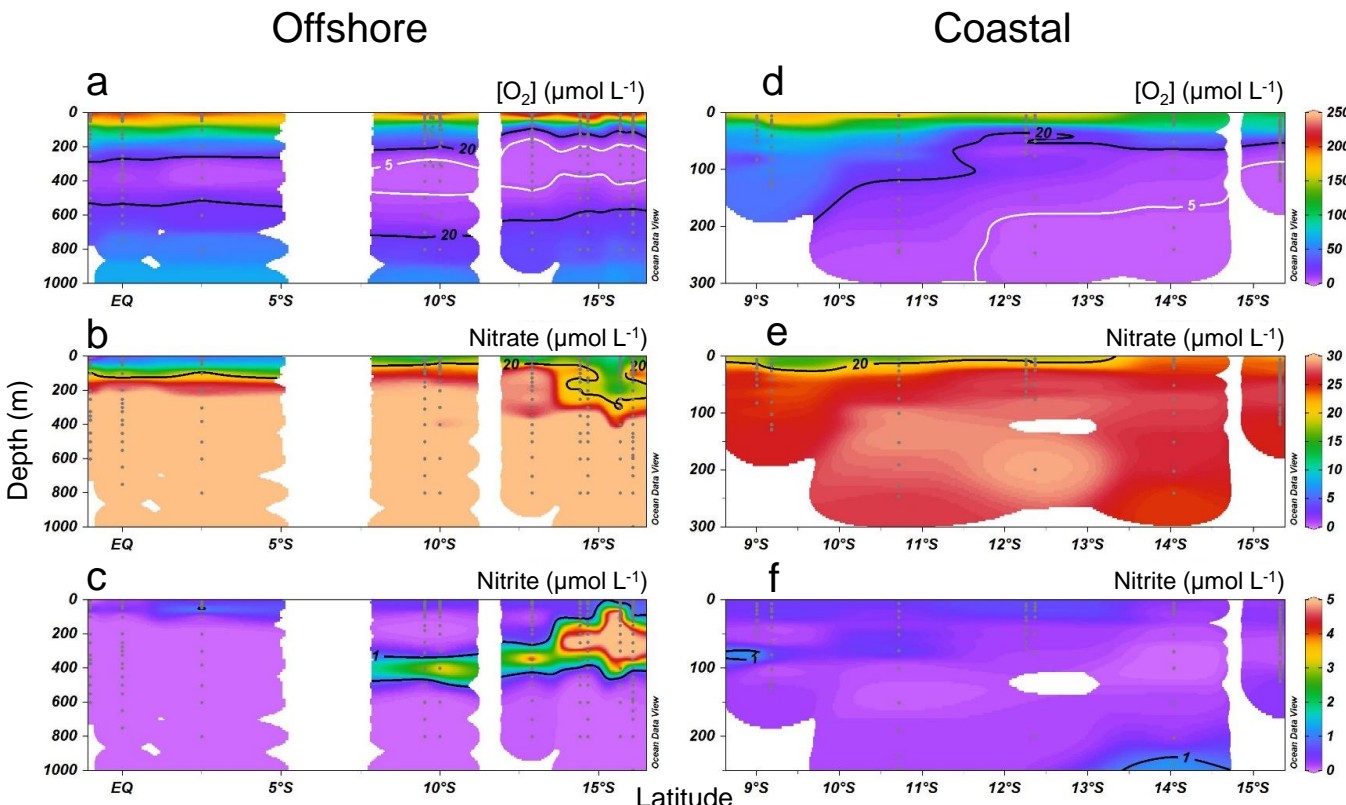

**Figure 3: Water column oxygen (a and d), nitrate (b and e) and nitrite concentrations (c and f) along the offshore (a,**
5  **b and c) and coastal sections (d, e and f) during October 2015.**



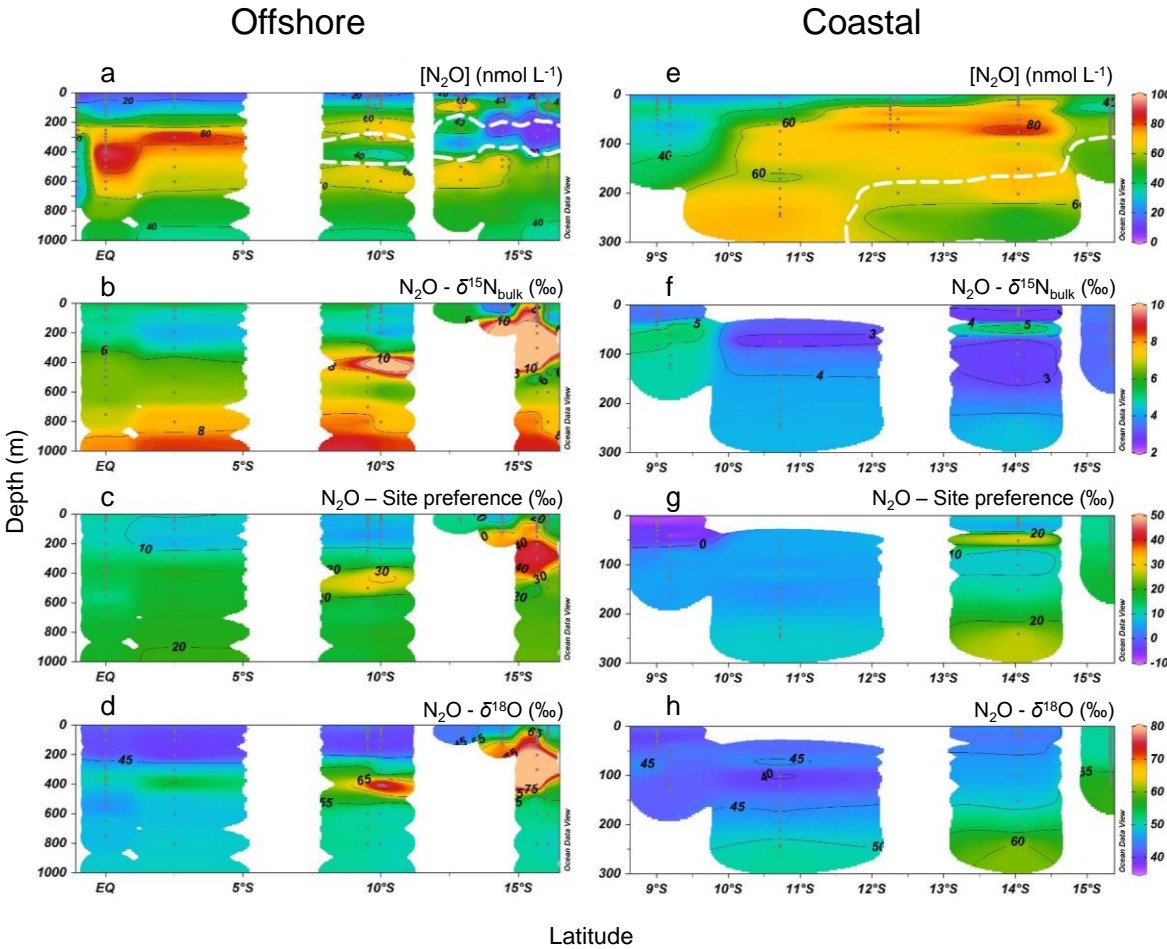

**Figure 4: Water column N₂O concentrations (a and e), δ¹⁵N_bulk (b and f), site preference (c and g) and δ¹⁸O (d and h) along the offshore (a, b, c and d) and coastal sections (e, f, g and h) during October 2015. White contour line in (a) and (e) denote the boundary of oxygen deficient zone ([O₂] = 5 μmol L⁻¹ isoline)**





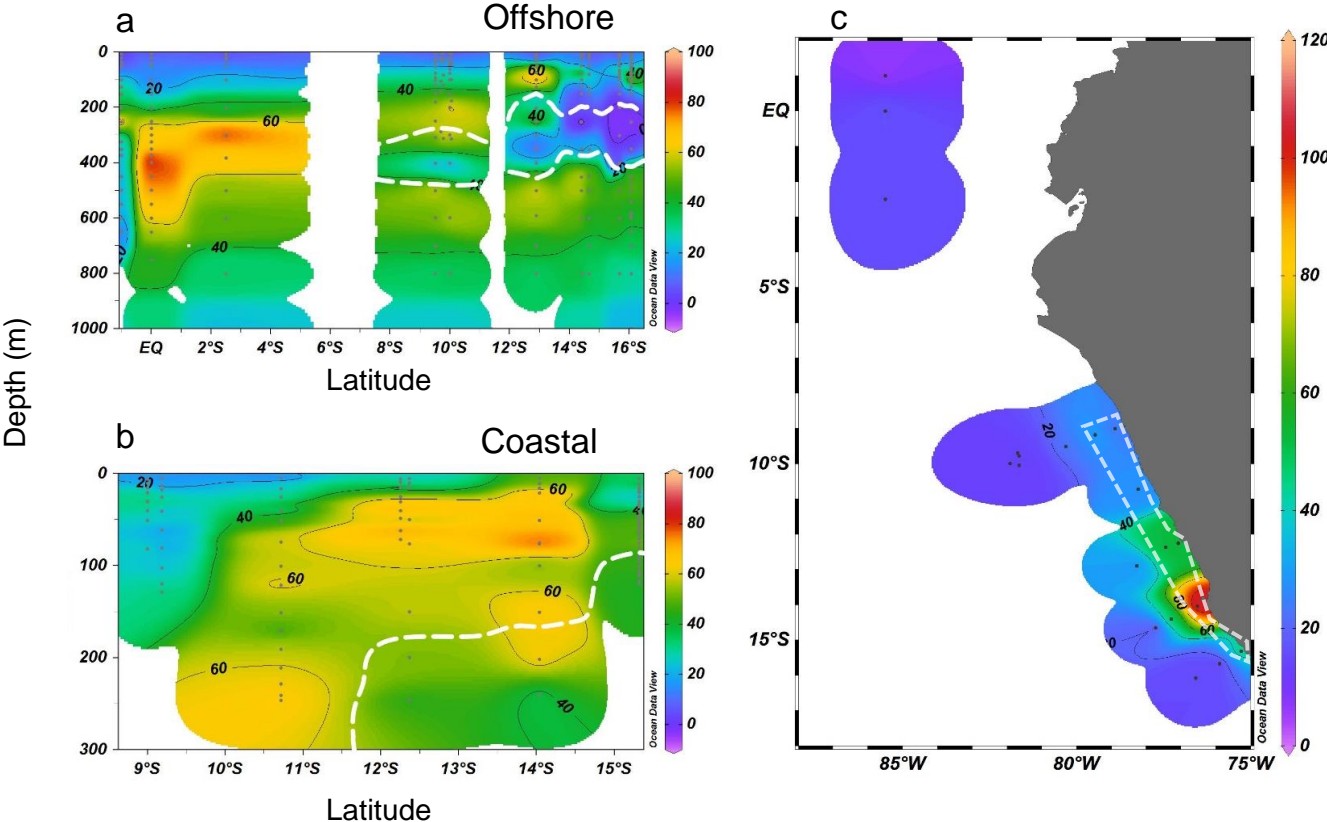

**Figure 5: N₂O excess (ΔN₂O, nmol L⁻¹) at the offshore section (a) and the coastal section (b) during October 2015; the white dashed line indicates the boundary of the oxygen deficient zone ([O₂] = 5 µmol L⁻¹ isoline). (c) Surface N₂O efflux (µmol m⁻² d⁻¹) from offshore and coastal stations (enclosed in white polygon) during October 2015.**





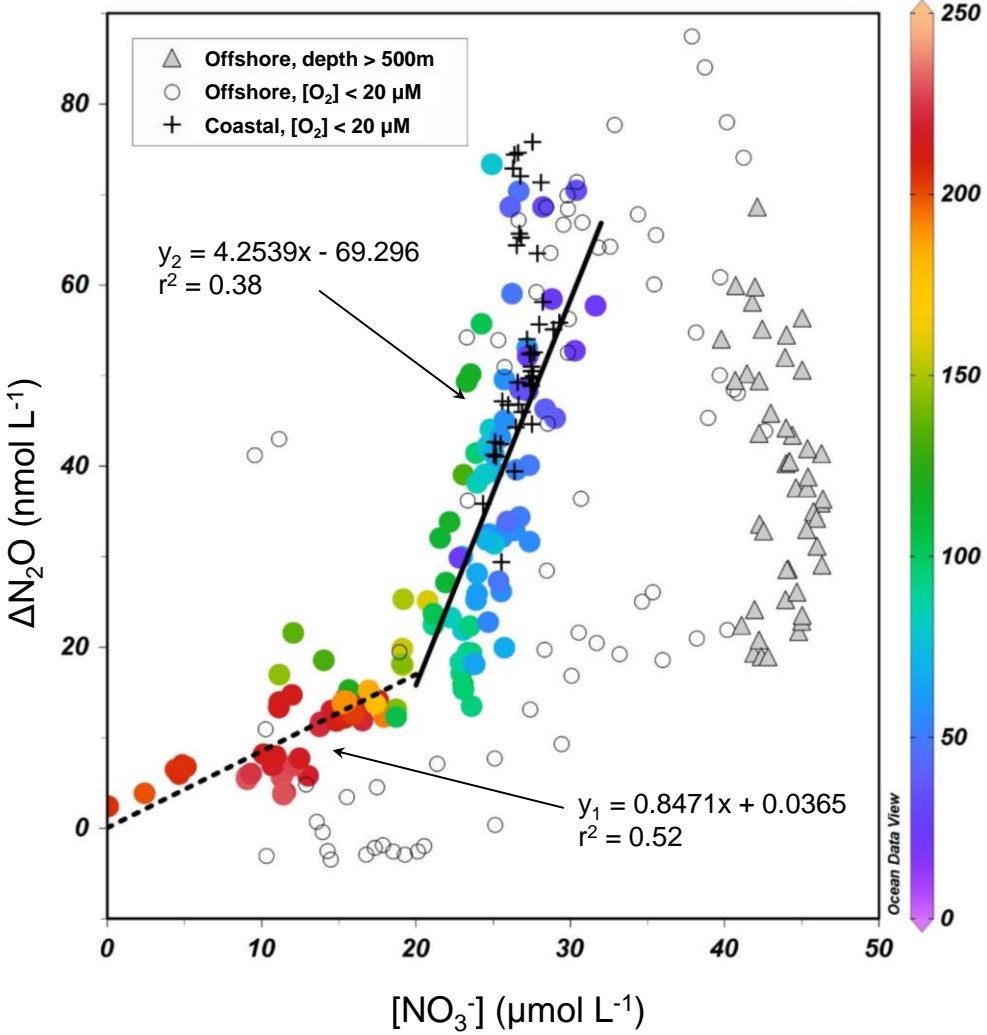

**Figure 6: NO$_3^-$-ΔN$_2$O relationship for samples from the upper oxycline ([O$_2$] > 20 µmol L$^{-1}$, depth < 500 m, colored circles), low oxygen ([O$_2$] < 20 µmol L$^{-1}$) coastal waters (+), low oxygen offshore waters (open circles), and the lower oxycline (depth > 500 m, filled triangles). Color bar shows the O$_2$ concentrations (µmol L$^{-1}$). For samples with NO$_3^-$ concentrations higher and lower than 20 µmol L$^{-1}$, two linear regressions were performed separately.**





**Figure 7: Surface ΔN₂O (a and d), meridional water column ΔN₂O distribution (b and e) and zonal water column ΔN₂O distribution (c and f) in October 2015 and in December 2012. Color bars for ΔN₂O (nmol L⁻¹) are shown in d, e and f. For meridional ΔN₂O distribution (b and e), data are from the coastal section, shown as white dashed polygon**

5    **in panel (a) and (d). For zonal ΔN₂O distribution (c and f), data are from a section 12 – 13°S, shown as white rectangle. In (b) and (e) the "20" contour line (black) denotes the [O₂] = 20 µmol L⁻¹ isoline, equivalent to the lower boundary of oxygenated layer; the "5" contour line (white) denotes the [O₂] = 5 µmol L⁻¹ isoline, equivalent to the upper boundary of oxygen deficient zone.**





**Figure 8: Depth profiles of N₂O concentration excess (ΔN₂O, nmol L⁻¹) measured at 6 different stations representing offshore (a, b and c) and coastal waters (d, e and f) during February 1985 (filled squares in f), January 2009 (filled triangles in e), October 2011 (filled diamonds in d), November 2012 (filled circles in a and b), December 2012 (open squares in c), January 2015 (open circles in e and f) and October 2015 (crosses). Profiles of 2015 are indicated in red and other years in blue.**





**Figure 9 Comparison of depth-integrated N₂O concentrations between El Niño (red bars) and normal years (blue bars). Station A, B and C are characterized as offshore stations whereas D,E and F are as coastal stations. See Figure 2a for station locations and Table S1 for references.**





**Table 1: Isotopic signature of produced N₂O estimated by linear regression of isotopomer ratios and inverse N₂O concentrations (see section 4.1 for model description and supplementary Figure S1 for results) in ODZ, N₂O peak, oxycline and surface layers.**

| Layer | Definition | | $\delta^{15}N_{bulk}$ (‰) | $\delta^{18}O$ (‰) | SP (‰) |
|---|---|---|---|---|---|
| Upper oxycline and surface | $[O_2] > 5\ \mu mol\ L^{-1}$  Depth < 500 m | Produced N₂O | 2.8 | 45.9 | 6.4 |
| | | Standard error | 0.3 | 1.2 | 1.9 |
| | | $R^2$ (n = 76) | 0.37 | 0 | 0.04 |
| N₂O peak | $[O_2] = 5 - 20\ \mu mol\ L^{-1}$  Depth < 500 m | Produced N₂O | 5.4 | 41.3 | 8.3 |
| | | Standard error | 0.9 | 3.0 | 3.0 |
| | | $R^2$ (n = 48) | 0.04 | 0.24 | 0.08 |
| Oxygen deficient zone | $[O_2] < 5\ \mu mol\ L^{-1}$  $[NO_2^-] > 1\ \mu mol\ L^{-1}$ | Produced N₂O | 8.5 | 71.0 | 39.9 |
| | | Standard error | 1.5 | 4.5 | 4.4 |
| | | $R^2$ (n = 11) | 0.38 | 0.40 | 0.01 |
| Intermediate waters | Depth = 500 - 1000 m | Produced N₂O | 3.6 | 50.0 | 15.6 |
| | | Standard error | 0.6 | 2.4 | 4.1 |
| | | $R^2$ (n = 21) | 0.69 | 0 | 0.04 |

**Acknowledgements**

The German Federal Ministry of Education and Research (BMBF) grant (03G0243A) awarded to C. Marandino, D. Grundle and T. Steinhoff supported the ASTRA-OMZ cruise onboard the R/V Sonne in October 2015. The Deutsche
10  Forschungsgemeinschaft (DFG) provided support via the Sonderforschungsbereich 754: Climate-Biogeochemistry Interactions in the Tropical Ocean and funded the R/V Meteor cruises. The BMBF also supported this study as part of the SOPRAN project I and II (03F0611A, 03F0662A). We thank the captains and crews of the R/V Meteor and R/V Sonne cruises for their help, C. Marandino (co-chief scientist) and T. Steinhoff for co-organizing the R/V Sonne cruise with D. Grundle, and M. Lohmann, H. Campen and M. Sun for the oxygen and nutrient measurements and help with N₂O sampling.
15  We thank the Peruvian authorities for allowing us to conduct work in their territorial waters. We thank Tina Baustian contributing hydrography and N₂O data off the Peruvian coast. In preparation of the manuscript, C. Marandino and L. Stramma provided constructive comments. Ji also received support from a DFG grant awarded to D. Grundle and C. Marandino.



**Data availability**

Raw data presented in this manuscript can be found in the Supplementary material.

**Author contributions**

DSG developed the experimental design. HWB, MIG, XM, DLA-M, DSG conducted field sampling. MA, XM, DLA-M conducted laboratory analyses. QJ and DSG perform data synthesis. QJ, MA, HWB, MIG, XM, DLA-M, DSG prepared the manuscript.

**Competing Interests**

The authors declare that they have no conflict of interest.

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
