# Peer review of "Investigating the effect of El Niño on nitrous oxide distribution in the Eastern Tropical South Pacific"

_Biogeosciences, 2018_

## Referee Comment (RC1) · Anonymous Referee #1 · 9 Nov 2018

This manuscript discusses nitrous oxide in the ETSP and the impact that El Nino has on this important trace gas. The authors have a lot of experience on this topic and consequently a high quality is expected of this manuscript. At the moment, this manuscript needs improving in a number of areas that are outlined below.

The biggest technical issue with this manuscript is that it covers two topics. The first topic is an overall expedition report from 2015 including measurements of N2O concentrations, fluxes, inventories, and isotopes. The second topic is an evaluation on the impact that El Nino has on N2O dynamics which necessitates an in-depth comparison of previous datasets. Both of these topics are very worthy of publication and the authors have the intellect and experience to document the new insights and perspectives gained from both topics. However, at the moment, I feel that 75% of the manuscript is

[Figure]

about the 2015 expedition and 25% is about an analysis of the effects of El Nino on N2O. For example, the comparison between 2015 and previous years is limited to the final section of the discussion. This is not consistent with the title of the manuscript which indicates to a reader that a more in-depth comparison will be provided. Its up to the authors whether they address this by providing a greater comparison with El Nino years or whether they save this for a later manuscript.

Another issue that I would like the authors to deal with is the absence of precision and accuracy values. The authors state that triplicate samples were collected, but no error bars are present on the vertical profiles and no values of analytical precision are provided. This is a problem when trying to compare measurements from separate years.

Specific comments

Line 16 – what region

Line 17. This sentence need re-writing

Line 18. I am not sure why you include this single summary sentence when in the discussion you highlight four water parcels with different pathways.

Line 20 level of sea surface N2O supersaturation. I understand what you mean, but a quick glance indicates you are talking about sea levels.

Line 25 Depth-integrated concentrations, change to water-column inventories?

page 2

Line 1 This sentence is a lazy description of El Nino, La Nina, and neutral. A schematic diagram would be great here to orientate the reader

Line 15, But what was the results of the modeling? Higher or lower nitrous oxide? Do your observations match the modeled predictions? An evaluation of El Nino on N2o would benefit greatly from the use of model predictions and I am not sure why the

authors did not leverage this information better

Line 10 Here you say ODZ (and do not spell out), while on the previous page you say OMZ

Line 15-18. The method needs to contain values of analytical precision and accuracy. This is particularly important for this study as you are comparing data from separate cruises, conducted several years apart. For example, Figure 8 does not have any error bars and how is the reader supposed to make an informed decision about differences between the separate expeditions.

Page 5.

Line 15 Why are concentrations reported moles per liter rather than moles per kg?

Line 23 This should be in the methods section, close to Equation 1, which is the equivalent calculation for N2O

Figure 1. Its not clear to me why you do not show the El Nino index against time and indicate along the timeline when the cruises were conducted. This would be easier to read that the current figure?

Figure 2a is an anomaly yet Figure 2b isn't. If you wanted to compare 2015 with other years you should show how the water masses vary with El Nino.

Figure 3. This Figure needs improving.

1 The units should be in moles per kg.

2. I am not sure you need to show depths of 500-1000 m since this takes up half the plot and is not discussed much in the text.

3. Please highlight the stations better.

4. I suggest you start other programs to make contour plots in the future (e.g. R) as

the ODV palette is not helpful for highlight the data trends that you have here.

Figure 4. I cannot see the individual points in the offshore stations so its hard to determine the extrapolation that has been applied

Figure 5. I am not sure this Figure adds value. Figure 5A and Figure5b are very similar to Figure 4 a and 4e (identical patterns, just different units. Figure 5c takes up a lot of space and only the bottom right hand section of the chart has any useful data.

Figure 6. You should connect this figure to the water masses identified in Table 1

Figure 8 With no error bars, it is not possible for the reader to know when there is a statistical difference between two depth profiles. On Page 3, Line 13 you say that triplicate measurements were taken, so they should be included

Table 1. This Table needs improving. 1. Please number the water parcels so they can be easily cross-referenced with the text. 2. Please report depth, O2, and nitrite concentrations for all four identified water parcels. 3. There is no column heading for the third column. 4 What is produced N2O?

---

## Referee Comment (RC2) · Anonymous Referee #3 · 27 Nov 2018

This manuscript presents the distribution and fluxes of N2O in the Eastern Tropical South Pacific region during Oct. 2015, when a strong El Nino event occurred. Measurements of N2O and other related parameters along with its isotopomers were made in the water samples collected from six stations. These measurements have been used to study the variability and biogeochemistry of N2O in the ocean water as well as the effect of this event on the distribution and fluxes of N2O in this region. The manuscript presents important results during this major El Nino event and it is very well written. However, I have the following clarifications/suggestions for its further improvement.

Specific points:

1. The main focus of this manuscript is on the effect of El Nino of the distribution and fluxes of N2O. The three offshore stations show buildup of N2O in the water down to

1000m depth (Fig. 8). However, the comparison for these 3 stations is limited with previous one neutral year only (2012). Also there is large variability in the 0-200m depth. Please show error bars for each point. Measurements for the three coastal stations are compared with the measurements from three different years (2011, 2009 and 1985). All these three stations show very different comparisons. Hence, it is difficult to conclude for the coastal region.

2. Fig. 9 shows depth integrated N2O concentrations and comparison with earlier measurements. However, the depth taken for each station is limited by earlier measurements and it is different for different stations except for stations B and C. This, in my view, is not correct and gives a wrong comparison. The X axis scale and even the depth for the coastal stations could have been same for all the three stations for a better visualization.

3. Are these earlier measurements for the same respective stations? If not, please give their locations also.

4. How the observed decrease in the N2O fluxes compare with earlier studies mentioned in the introduction (P2, L17)?

5. P1, L25 : 'The depth-integrated N2O….were nearly twice . . ..' is not correct except may be for the E and F stations. Please modify this sentence suitably and also give depth information related to integration.

6. How long this El Nino event has been there? The ONI shown in Fig. 1 for 2015 was >0.5 in January itself.

Minor corrections:

P1, L16: '….was developing ..' or developed? P2, L17: Please change to – '…related to changes in. . .'
* * *

---

## Referee Comment (RC3) · Anonymous Referee #4 · 12 Jan 2019

p. 1 lines 19-24: The deepening of the oxycline and deeper N2O peaks—is this just due to a deepening of the isopycnal they occur on, or are they shifting to different density surfaces?

p. 3, lines 19-25: Additional detail on analysis and calibration of N2O isotopologue measurements should be provided.

p. 4, Equation 5: definition of SP should not have the multiple of 1/2.

p. 6, lines 17-18: how do you interpret the SP values < 0, especially in the surface which should likely be closer to atmospheric values ($\sim$18‰?

p. 7, equation 6: I understand that you are using an end member mixing, or 'Keeling Plot' model for estimating the d value of the produced N2O, but it was less clear how

you then attribute the N2O source based on those values. Please provide additional information, support, justification for the source attributions based on your measurements and calculations.

p. 7, lines 27-28: it seems difficult to explain a SP value of -9‰ under these conditions. I would be concerned here about calibration, and believe that the authors should further discuss their calibration techniques, and possible explanations for such a low SP value in surface waters.

p. 8, line 1: Is there any other evidence of upwelling/diffusion from suboxic zones influencing the upper water column here?

p. 8, lines 19-22: It is not meaningful to apply the model in equation 6 to an environment in which N2O is being consumed. Is there any evidence of N2O production here, or only consumption?

p. 8, line 28: Are you excluding nitrifier-denitrification here, or lumping it in with 'denitrification'?

p. 9, lines 1-3: How would you derive N2O production rates from nitrate and nitrite isotopes?

p. 10: I thought the discussion of El Nino effects on N2O was interesting, and wondered to what extent the sampling period represented conditions during the 2015 El Nino event, and how the 2015 El Nino event might reflect other El Nino events in this region. How broadly applicable are the current results?

Figure 1: Why are these years/data in particular the ones compared in this figure? It seems more relevant to compare the strength of the 2015 to other strong El Nino years to gauge how representative were the conditions sampled here of other El Nino periods.

Figures 3, 4, 5, 7: These section plots are not a particularly effective way to present the data. I understand the desire to present the data in a spatially explicit fashion, but the

section plots, with only a few stations, rather large gaps in the data, and considerable smoothing don't help to relay the information to readers. In many of these cases, stacked depth profiles would be a much clearer way of presenting the patterns and comparing to other existing data.

---

## Author Comment (AC1) · 1 Feb 2019

Anonymous Referee #1 [Referee] This manuscript discusses nitrous oxide in the ETSP and the impact that El Nino has on this important trace gas. The authors have a lot of experience on this topic and consequently a high quality is expected of this manuscript. At the moment, this manuscript needs improving in a number of areas that are outlined below.

[Referee] The biggest technical issue with this manuscript is that it covers two topics. The first topic is an overall expedition report from 2015 including measurements of N2O concentrations, fluxes, inventories, and isotopes. The second topic is an evalua-tion on the impact that El Nino has on N2O dynamics which necessitates an in-depth

comparison of previous datasets. Both of these topics are very worthy of publication and the authors have the intellect and experience to document the new insights and perspectives gained from both topics. However, at the moment, I feel that 75% of the manuscript is about the 2015 expedition and 25% is about an analysis of the effects of El Nino on N2O. For example, the comparison between 2015 and previous years is limited to the final section of the discussion. This is not consistent with the title of the manuscript which indicates to a reader that a more in-depth comparison will be provided. It's up to the authors whether they address this by providing a greater comparison with El Nino years or whether they save this for a later manuscript.

[Response] We agree with the reviewer that the manuscript has two major components: an overall expedition report and a comparative study of El Nino effects on N2O dynamics. We think the two are closely interlinked and cannot be separated into two topics or even two manuscripts. We disagree with the reviewer's point about "75% of the manuscript is about the 2015 expedition and 25% is about an analysis of the effects of El Nino on N2O"; it is necessary to first introduce methods and dataset (N2O isotopes, substrates availabilities and temporal dynamics), then move on to comprehensive presentation of the effects of El Nino on N2O dynamics, which is in line with the title. The current dataset suggests that change of water column physical structure explains the N2O depth distribution and flux dynamics during a strong El Nino event. We also observed, for the first time, higher water column N2O inventory during El Nino. We will improve the clarity of the manuscript, e.g. describing the main contents of the manuscript in the last paragraph of the Introduction section, so readers can navigate the manuscript more easily.

[Referee] Another issue that I would like the authors to deal with is the absence of precision and accuracy values. The authors state that triplicate samples were collected, but no error bars are present on the vertical profiles and no values of analytical precision are provided. This is a problem when trying to compare measurements from separate years.

[Response] Good point. The typical analytical precision of N2O concentration measurement during 2015 cruise is < 2 nmol/L; and the precision from previous cruises were generally < 5%, which we will add onto the figures in the revised manuscript. The precision of our measurements are lower than El Nino variability and we think the conclusion of higher water column N2O inventories during El Nino will hold.

Specific comments [Referee] Line 16 – what region

[Response] This region refers to the Eastern Tropical South Pacific (ETSP)

[Referee] Line 17. This sentence need re-writing

[Response] We rewrite the sentence as follows: "In October 2015, a strong El Niño event was developing in the ETSP; we conduct field observation to investigate (1) the N2O production pathways and (2) the effects of El Niño on water column N2O distributions and fluxes using data from previous non-El Niño years."

[Referee] Line 18. I am not sure why you include this single summary sentence when in the discussion you highlight four water parcels with different pathways.

[Response] We included this sentence because we want to highlight that both nitrification and denitrification are contributing N2O production in the near surface waters, which contribute to effluxes to the atmosphere.

[Referee] Line 20 level of sea surface N2O supersaturation. I understand what you mean, but a quick glance indicates you are talking about sea levels.

[Response] We rewrite the sentence as follows: "Higher than normal sea surface temperatures were associated with a deepening of the oxycline. Within the shelf region, surface N2O supersaturation was nearly an order of magnitude lower than those of non-El Niño years."

[Referee] Line 25 Depth-integrated concentrations, change to water-column inventories?

**BGD**

[Response] Yes. We rewrite part of the sentence as follows: "Water-column inventories of N2O within the top 1000 m were 0 – 160% higher than those measured in non-El Niño years,"

page 2 [Referee] Line 1 This sentence is a lazy description of El Nino, La Nina, and neutral. A schematic diagram would be great here to orientate the reader

[Response] This sentence points out the most contrasting feature of El Nino vs. La Nina events, and we don't think a schematic diagram is needed.

[Referee] Line 15, But what was the results of the modeling? Higher or lower nitrous oxide? Do your observations match the modeled predictions? An evaluation of El Nino on N2O would benefit greatly from the use of model predictions and I am not sure why the authors did not leverage this information better

[Response] The modelling studies showed lower water column denitrification (Yang et al., 2017, GBC) and higher nitrification, lower N2O fluxes (Mogollón and Calil, 2017) during El Nino. The citation of Carrasco et al. is a typo. We will briefly introduce these model results in the revised text.

Page 3 [Referee] Line 10 Here you say ODZ (and do not spell out), while on the previous page you say OMZ

[Response] The ODZ refers to 'oxygen deficient zone', which is defined as dissolved < 5 micromoles per liter. We added the full name in the revised manuscript.

[Referee] Line 15-18. The method needs to contain values of analytical precision and accuracy. This is particularly important for this study as you are comparing data from separate cruises, conducted several years apart. For example, Figure 8 does not have any error bars and how is the reader supposed to make an informed decision about differences between the separate expeditions.

[Response] Good point. The analytical precision of N2O concentration measurement is < 2 nmol/L, and we will add the error bars for 2015 data on Figure 8. The precision of previous dataset was generally < 5%. The precision of the measurements are lower than El Nino variability and we think the conclusion of higher water column N2O inventories during El Nino will hold.

Page 5. [Referee] Line 15 Why are concentrations reported moles per liter rather than moles per kg?

[Response] We chose to report all the concentrations in moles per liter because of consistency within this manuscript, and with earlier dataset that employed the same unit.

[Referee] Line 23 This should be in the methods section, close to Equation 1, which is the equivalent calculation for N2O

[Response] Indeed. We will rework the sentence at the revised manuscript.

[Referee] Figure 1. Its not clear to me why you do not show the El Nino index against time and indicate along the timeline when the cruises were conducted. This would be easier to read that the current figure?

[Response] It is a good suggestion. We tried to plot as the reviewer suggested, and realized there was a large gap between 1985 and 2009 with no cruise being conducted. We think the current presentation is informative because it clearly shows the contrasting feature of 2015 El Nino comparing to the rest of periods.

[Referee] Figure 2a is an anomaly yet Figure 2b isn't. If you wanted to compare 2015 with other years you should show how the water masses vary with El Nino.

[Response] We compared the T-S diagram between Oct. 2015 and Nov. 2012; the main difference was shallow water was warmer in Oct. 2015. Below the thermocline, water masses are apparently similar. We will add a panel to Figure 2 in the revised text. Figure T-S diagram of November 2012 (left) and October 2015 (right)

[Referee] Figure 3. This Figure needs improving. 1 The units should be in moles per

kg. 2. I am not sure you need to show depths of 500-1000 m since this takes up half the plot and is not discussed much in the text. 3. Please highlight the stations better. 4. I suggest you start other programs to make contour plots in the future (e.g. R) as the ODV palette is not helpful for highlight the data trends that you have here.

[Response] To be consistent with previous measurements, we will keep the unit, mole per liter for chemical constituents. We will keep the offshore data from 0 – 1000 m because the El Nino effect is thought to occur in the upper 1000m, and the discussion (Figure 7 and 8) compared the top 1000 m of water column properties.

[Referee] Figure 4. I cannot see the individual points in the offshore stations so it's hard to determine the extrapolation that has been applied

[Response] We'll increase the contrast of the data points in the revised manuscript.

[Referee] Figure 5. I am not sure this Figure adds value. Figure 5A and Figure5b are very similar to Figure 4a and 4e (identical patterns, just different units. Figure 5c takes up a lot of space and only the bottom right hand section of the chart has any useful data.

[Response] Figure 5 shows N2O excess and associated surface fluxes. From there we can see that although surfaces fluxes are low, water column is still oversaturated with N2O, suggesting change of hydrography. Our feeling is that it will be more confusing for the readers not to show Figure 5.

[Referee] Figure 6. You should connect this figure to the water masses identified in Table 1

[Response] Good point. Some water masses (oxycline and offshore > 500 m samples) are identified in Table 1. We will add some clarifying sentences in section 4.2.

[Referee] Figure 8. With no error bars, it is not possible for the reader to know when there is a statistical difference between two depth profiles. On Page 3, Line 13 you say that triplicate measurements were taken, so they should be included.

[Response] Good point. The analytical precision of N2O concentration measurement is < 2 nmol/L, and we will add the error bars for 2015 data on the plot. The precision of previous dataset was generally < 5%.

[Referee] Table 1. This Table needs improving. 1. Please number the water parcels so they can be easily cross-referenced with the text. 2. Please report depth, O2, and nitrite concentrations for all four identified water parcels. 3. There is no column heading for the third column. 4 What is produced N2O?

[Response] We will number the water parcels so that readers can cross-reference in the text. We will report depth, oxygen and nitrite concentrations for all water parcels. The heading of third column should be "statistical properties". The term "Produced N2O" is defined in the first paragraph of section 4.1, as "isotopic signature of N2O produced within the water mass".

Please also note the supplement to this comment:
https://www.biogeosciences-discuss.net/bg-2018-453/bg-2018-453-AC1-supplement.pdf

[Figure]

**Fig. 1.**

---

## Author Comment (AC2) · 1 Feb 2019

[Referee] This manuscript presents the distribution and fluxes of N2O in the Eastern Tropical South Pacific region during Oct. 2015, when a strong El Nino event occurred. Measurements of N2O and other related parameters along with its isotopomers were made in the water samples collected from six stations. These measurements have been used to study the variability and biogeochemistry of N2O in the ocean water as well as the effect of this event on the distribution and fluxes of N2O in this region. The manuscript presents important results during this major El Nino event and it is very well written.

However, I have the following clarifications/suggestions for its further improvement.

[Figure]

Specific points:

[Referee] 1. The main focus of this manuscript is on the effect of El Nino of the distribution and fluxes of N2O. The three offshore stations show buildup of N2O in the water down to 1000m depth (Fig. 8). However, the comparison for these 3 stations is limited with previous one neutral year only (2012). Also there is large variability in the 0-200m depth. Please show error bars for each point. Measurements for the three coastal stations are compared with the measurements from three different years (2011, 2009 and 1985). All these three stations show very different comparisons. Hence, it is difficult to conclude for the coastal region.

[Response] The data availability allowed us to compare offshore water column N2O inventories during between 2015 and 2012. It is the scope of this paper to compare water column properties during El Nino vs. non-El Nino years. In coastal waters, the water column inventories were significantly higher (15 – 160% higher) during El Nino times. These apparent evidence led us to conclude that water column N2O inventories at lower latitudes during El Nino years were higher than those during non El Nino years. The analytical precision of N2O concentration measurement is < 2 nmol/L, and we will add the error bars for 2015 data on the plot. The precision of previous dataset was generally < 5%. The precision of our measurements are lower than El Nino variability and we think the conclusion of higher water column N2O inventories during El Nino will hold.

[Referee] 2. Fig. 9 shows depth integrated N2O concentrations and comparison with earlier measurements. However, the depth taken for each station is limited by earlier measurements and it is different for different stations except for stations B and C. This, in my view, is not correct and gives a wrong comparison. The X axis scale and even the depth for the coastal stations could have been same for all the three stations for a better visualization.

[Response] We compared the integrated N2O at the same depth range for each station;

and in the revised manuscript, the depth range for offshore waters will be 0 – 800. For coastal waters, the range is shallower than 300 meters. The effects of El Nino are generally thought to be confined in the thermocline, and thus we don't expect significant changes below 1000 m at offshore waters. For coastal waters, the water depth is generally shallower than 300 m; in some cases, the entire water column was effected by El Nino. We will clarify this section in the revised draft.

[Referee] 3. Are these earlier measurements for the same respective stations? If not, please give their locations also.

[Response] All the location info for the data presented in Figure 9 are presented in the supplementary material Table S1. Although some measurements were not made at the exact location, data are comparable when measurements were made within 0.75 by 0.75 degree grid.

[Referee] 4. How the observed decrease in the N2O fluxes compare with earlier studies mentioned in the introduction (P2, L17)?

[Response] Observation from 2015 – 16 El Nino event showed 23 – 108 $\mu$mol/m2/d, 75 – 95 % reduction of fluxes of December 2012 (459 – 1825 $\mu$mol/m2/d). This is consistent with observation from Cline et al. (1987) who reported 80% reduction in fluxes.

[Referee] 5. P1, L25 : 'The depth-integrated N2O....were nearly twice....' is not correct except may be for the E and F stations. Please modify this sentence suitably and also give depth information related to integration.

[Response] We rewrite part of the sentence as follows: "Water-column inventories of N2O within the top 1000 m were 0 – 160% higher than those measured in non-El Niño years,"

[Referee] 6. How long this El Nino event has been there? The ONI shown in Fig. 1 for 2015 was >0.5 in January itself.

[Response] Given the definition of El Nino event being ONI > 0.5, the event started in November 2014 and lasted until May 2016. We will include this information in the Introduction section.

Minor corrections: [Referee] P1, L16: '....was developing ..' or developed?

[Response] The El Nino event was still developing in Oct. 2015, as indicated by ONI in Figure 1

[Referee] P2, L17: Please change to – '...related to changes in...'

[Response] Done

Please also note the supplement to this comment:
https://www.biogeosciences-discuss.net/bg-2018-453/bg-2018-453-AC2-supplement.pdf

---

## Author Comment (AC3) · 1 Feb 2019

Anonymous Referee #4 [Referee] p.1 lines 19-24: The deepening of the oxycline and deeper N2O peaks is this just due to a deepening of the isopycnal they occur on, or are they shifting to different density surfaces?

[Response] Our data suggest the deepening of the oxycline and deeper N2O peaks are more likely due to deepening the isopycnal they occur on. We'll make it clear in the Abstract.

[Referee] p.3, lines 19-25: Additional detail on analysis and calibration of N2O isotopologue measurements should be provided.

[Response] We added the following in the revised manuscript: The mass ratio of 45/44

and 46/44 were used to derive $\delta15Nbulk$-N2O and $\delta18O$-N2O and, respectively; while the mass ratio 31/30 was used to derive $\delta15N\alpha$-N2O. Then $\delta15N\beta$-N2O was calculated from equation (4) using $\delta15Nbulk$-N2O and $\delta15N\alpha$-N2O values. Calibration of $\delta15N\alpha$-N2O, $\delta15N\beta$-N2O and $\delta18O$-N2O was accomplished using 4 certified standard gases (supplied by Joachim Mohn) encompassing the values reported here. The analytical precision of isotope measurements were $\pm0.07$, 0.17, 0.36 and 0.18‰ for $\delta15Nbulk$-N2O, $\delta15N\alpha$-N2O, $\delta15N\beta$-N2O and $\delta18O$-N2O, respectively.

[Referee] p.4, Equation 5: definition of SP should not have the multiple of 1/2.

[Response] Indeed. Corrected.

[Referee] p.6, lines 17-18: how do you interpret the SP values < 0, especially in the surface which should likely be closer to atmospheric values (18‰?

[Response] We checked the analysis and we are confident about the measurements. We interpreted as the denitrification dominant N2O source. Some recent publications showed that N2O produced by bacterial denitrification has SP values ranging from 0 to minus 10 per mil. Thus we attribute our measurements to denitrification in the water column. We will add these explanation in the revised text. Mothet et al., 2013: http://dx.doi.org/10.1071/EN13021 Winther et al., 2018: https://www.biogeosciences.net/15/767/2018/bg-15-767-2018.pdf

[Referee] p.7, equation 6: I understand that you are using an end member mixing, or 'Keeling Plot' model for estimating the d value of the produced N2O, but it was less clear how you then attribute the N2O source based on those values. Please provide additional information, support, justification for the source attributions based on your measurements and calculations.

[Response] N2O production pathways can be inferred from SP, because SP is independent of isotopic values of N2O production substrates; generally, N2O produced via NH4+ oxidation and partial denitrification have distinctive SP values of 30 ± 5 ‰

and $0 \pm 5$ ‰ respectively. Simply put, higher produced SP value indicates nitrification whereas lower SP indicates denitrification. It is supposed to be a qualitative, not a quantitative indicator. This is stated immediately followed equation 6 and cited Toyoda et al., 2011. The model and calculation is documented by Fuiji et al., 2013 (doi:10.1007/s10872-012-0162-4).

[Referee] p.7, lines 27-28: it seems difficult to explain a SP value of -9‰ under these conditions. I would be concerned here about calibration, and believe that the authors should further discuss their calibration techniques, and possible explanations for such a low SP value in surface waters.

[Response] We double checked the isotopic data; our in-house standard (cold seawater) and isotopic references showed no sign of inaccuracies. We are confident that the negative SP values are accurately measured. Some recent publications showed that N2O produced by bacterial denitrification has SP values ranging from 0 to -10 per mil. Thus we attribute our measurements to denitrification in the water column. We will add these explanation in the revised text. Mothet et al., 2013: http://dx.doi.org/10.1071/EN13021 Winther et al., 2018: https://www.biogeosciences.net/15/767/2018/bg-15-767-2018.pdf

[Referee] p.8, line 1: Is there any other evidence of upwelling/diffusion from suboxic zones influencing the upper water column here?

[Response] Diffusion can occur simply due to a concentration gradient; as the N2O concentration gradient near the concentration peak can be $3 - 5$ nmol/L/m. We will clarify the sentence here in the revised manuscript.

[Referee] p. 8, lines 19-22: It is not meaningful to apply the model in equation 6 to an environment in which N2O is being consumed. Is there any evidence of N2O production here, or only consumption?

[Response] This section requires more clarification. In the oxygen deficient zone

where N2O is undersaturated, N2O is being produced and consumed at the same time and reaching a dynamic balance. This is demonstrated by tracer incubation studies (Ji et al., 2015 GRL, doi:10.1002/2015GL066853; Babbin et al., 2015, doi:10.1126/science.aaa8380). The net isotope effects during N2O production and consumption by denitrification is quite significant that both N and O isotopic compositions increased. Therefore it is valid to apply equation 6 to distinguish N2O production.

[Referee] p.8, line 28: Are you excluding nitrifier-denitrification here, or lumping it in with 'denitrification'?

[Response] Because nitrifier-denitrification and denitrifier-denitrification have similar SP values, we use 'denitrification' to represent both possible pathways in this sentence.

[Referee] p.9, lines 1-3: How would you derive N2O production rates from nitrate and nitrite isotopes?

[Response] Details can be found in a previous study from Bourbonnais et al., 2017 (doi:10.1002/2016GB005567), in which the authors applied a three-dimensional model to derive N2O production rates from isotopic compositions of N2O, nitrate and nitrite, and fractionation factors during nitrification and denitrification.

[Referee] p.10: I thought the discussion of El Nino effects on N2O was interesting, and wondered to what extent the sampling period represented conditions during the 2015 El Nino event, and how the 2015 El Nino event might reflect other El Nino events in this region. How broadly applicable are the current results?

[Response] The cruise in October 2015 was conducted during a developing strong El Nino since early 2015 (see Figure 1). We realized that our study can be treated as a 'snapshot' during El Nino, rather than a continuous record of it. Previous studies (e.g. Stamma et al., 2016, doi:10.5194/os-12-861-2016) have concluded that 2015/16 El Nino can be categorized as "Strong El Nino" based on ONI and other properties. Thus we present comparative study of N2O dynamics between El Nino and non-El

Nino years.

[Referee] Figure 1: Why are these years/data in particular the ones compared in this figure? It seems more relevant to compare the strength of the 2015 to other strong El Nino years to gauge how representative were the conditions sampled here of other El Nino periods.

[Response] Previous studies (e.g. Stamma et al., 2016) have concluded that 2015/16 El Nino can be categorized as "Strong El Nino" based on ONI and other physical properties. Unfortunately, no N2O concentration or isotope data existed in the database for us to compare the N2O dynamics between two strong El Nino events.

[Referee] Figures 3, 4, 5, 7: These section plots are not a particularly effective way to present the data. I understand the desire to present the data in a spatially explicit fashion, but the section plots, with only a few stations, rather large gaps in the data, and considerable smoothing don't help to relay the information to readers. In many of these cases, stacked depth profiles would be a much clearer way of presenting the patterns and comparing to other existing data.

[Response] The authors have tried a number of ways plotting the data, including stacked depth profiles, to present the water column profiles during 2015/16 El Nino. And the current presentation is rather realistic and effective in conveying information. We realized in some cases there's a large gap between stations, so we leave some blank spaces; we realized the smoothing exerted by ODV software and we chose to use "weighted-averaged gridding" instead of the more popular "DIVA gridding"

---

## Author Response (AR1)

**Response to reviewers**

Please refer to colored text in the resubmitted manuscript for revisions.

**Anonymous Referee #1**

[Referee] This manuscript discusses nitrous oxide in the ETSP and the impact that El Nino has on this important trace gas. The authors have a lot of experience on this topic and consequently a high quality is expected of this manuscript. At the moment, this manuscript needs improving in a number of areas that are outlined below.

[Referee] The biggest technical issue with this manuscript is that it covers two topics. The first topic is an overall expedition report from 2015 including measurements of N2O concentrations, fluxes, inventories, and isotopes. The second topic is an evaluation on the impact that El Nino has on $N_2O$ dynamics which necessitates an in-depth comparison of previous datasets. Both of these topics are very worthy of publication and the authors have the intellect and experience to document the new insights and perspectives gained from both topics. However, at the moment, I feel that 75% of the manuscript is about the 2015 expedition and 25% is about an analysis of the effects of El Nino on $N_2O$. For example, the comparison between 2015 and previous years is limited to the final section of the discussion. This is not consistent with the title of the manuscript which indicates to a reader that a more in-depth comparison will be provided. It's up to the authors whether they address this by providing a greater comparison with El Nino years or whether they save this for a later manuscript.

*[Response] We agree with the reviewer that the manuscript has two major components: an overall expedition report and a comparative study of El Nino effects on $N_2O$ dynamics. We think the two are closely interlinked and cannot be separated into two topics or even two manuscripts. We disagree with the reviewer's point about "75% of the manuscript is about the 2015 expedition and 25% is about an analysis of the effects of El Nino on $N_2O$"; it is necessary to first introduce methods and dataset ($N_2O$ isotopes, substrates availabilities and temporal dynamics), then move on to comprehensive presentation of the effects of El Nino on $N_2O$ dynamics, which is in line with the title. The current dataset suggests that change of water column physical structure explains the $N_2O$ depth distribution and flux dynamics during a strong El Nino event. We also observed, for the first time, higher water column $N_2O$ inventory during El Nino. We improve the clarity of the manuscript, e.g. describing the main contents of the manuscript in page 1, line 23 - 29, so readers can navigate the manuscript more easily.*

[Referee] Another issue that I would like the authors to deal with is the absence of precision and accuracy values. The authors state that triplicate samples were collected, but no error bars are present on the vertical profiles and no values of analytical precision are provided. This is a problem when trying to compare measurements from separate years.

*[Response] Good point. The typical analytical precision of $N_2O$ concentration measurement during 2015 cruise is < 2.5 nmol/L, stated in page 3 line 28; and we assume 5% of the mean $N_2O$ concentrations as the precision from previous cruises, stated in page 5, line 10 − 11. We added the error bars onto Figure 8 and 9 in the revised manuscript. The precision of our measurements are lower than El Nino variability and we think the conclusion of higher water column $N_2O$ inventories during El Nino will hold.*

Specific comments
[Referee] Line 16 – what region

*[Response] This region refers to the Eastern Tropical South Pacific (ETSP)*

[Referee] Line 17. This sentence need re-writing
*[Response] We rewrite the sentence in page 1, line 16 – 19 as follows: "In October 2015, a strong El Niño event was developing in the ETSP; we conduct field observation to investigate (1) the N₂O production pathways and associated biogeochemical properties, and (2) the effects of El Niño on water column N₂O distributions and fluxes using data from previous non-El Niño years."*

[Referee] Line 18. I am not sure why you include this single summary sentence when in the discussion you highlight four water parcels with different pathways.
*[Response] We included this sentence because we want to highlight that both nitrification and denitrification are contributing N₂O production in the near surface waters, which contribute to effluxes to the atmosphere. Revised sentence in page 1 line 19 – 21.*

[Referee] Line 20 level of sea surface N2O supersaturation. I understand what you mean, but a quick glance indicates you are talking about sea levels.
*[Response] We rewrite the sentence in page 1, line 21 – 23: "Higher than normal sea surface temperatures were associated with a deepening of the oxycline and the oxygen minimum layer. Within the shelf region, surface N₂O supersaturation was nearly an order of magnitude lower than those of non-El Niño years."*

[Referee] Line 25 Depth-integrated concentrations, change to water-column inventories?
*[Response] We rewrite part of the sentence in page 1, line 26 – 27: "Water-column inventories of N₂O within the top 1000 m were up to 160% higher than those measured in non-El Niño years,"*

page 2
[Referee] Line 1 This sentence is a lazy description of El Nino, La Nina, and neutral. A schematic diagram would be great here to orientate the reader
*[Response] This sentence points out the most contrasting feature of El Nino vs. La Nina events, and we don't think a schematic diagram is needed.*

[Referee] Line 15, But what was the results of the modeling? Higher or lower nitrous oxide? Do your observations match the modeled predictions? An evaluation of El Nino on N2O would benefit greatly from the use of model predictions and I am not sure why the authors did not leverage this information better
*[Response] The modelling studies showed lower water column denitrification (Yang et al., 2017, GBC) and higher nitrification, lower N₂O fluxes (Mogollón and Calil, 2017) during El Nino. The citation of Carrasco et al. is a typo. We briefly introduce these model results and change the statement in page 2, line 18 – 21.*

[Referee] Line 10 Here you say ODZ (and do not spell out), while on the previous page you say OMZ
*[Response] The ODZ refers to 'oxygen deficient zone', which is defined as dissolved < 5 micromoles per liter. We added the full name in page 3 line 17.*

[Referee] Line 15-18. The method needs to contain values of analytical precision and accuracy. This is particularly important for this study as you are comparing data from separate cruises, conducted several years apart. For example, Figure 8 does not have any error

bars and how is the reader supposed to make an informed decision about differences between the separate expeditions.

*[Response] Good point. The typical analytical precision of N2O concentration measurement during 2015 cruise is < 2.5 nmol/L, stated in page 3 line 28; and we assume 5% of the mean N₂O concentrations as the precision from previous cruises, stated in page 5, line 10 – 11. We added the error bars onto Figure 8 and 9 in the revised manuscript. The precision of our measurements are lower than El Nino variability and we think the conclusion of higher water column N₂O inventories during El Nino will hold.*

Page 5.

[Referee] Line 15 Why are concentrations reported moles per liter rather than moles per kg?

*[Response] We chose to report all the concentrations in moles per liter because of consistency within this manuscript, and with earlier dataset that employed the same unit.*

[Referee] Line 23 This should be in the methods section, close to Equation 1, which is the equivalent calculation for N2O

*[Response] Indeed. This part is now in page 3, line 19.*

[Referee] Figure 1. Its not clear to me why you do not show the El Nino index against time and indicate along the timeline when the cruises were conducted. This would be easier to read that the current figure?

*[Response] It is a good suggestion. We tried to plot as the reviewer suggested, and realized there will be a large gap between 1985 and 2009 with no cruise being conducted. We think the current presentation is informative because it clearly shows the contrasting feature of 2015 El Nino comparing to the rest of periods.*

[Referee] Figure 2a is an anomaly yet Figure 2b isn't. If you wanted to compare 2015 with other years you should show how the water masses vary with El Nino.

*[Response] We compared the T-S diagram between Oct. 2015 and Nov. 2012; the main difference was shallow water was warmer in Oct. 2015. Below the thermocline, water masses are apparently similar. We add the following sentence in page 5, line 25 – 27: "The October 2015 water column below 250 m had similar thermohaline properties compared to those of October – December 2012 (non-El Niño) that had been shown in an earlier study (Kock et al., 2016), except that October 2015 had warmer surface water." Since the 2012 T-S diagram was presented in Kock et al., 2016, we will not add the comparison on Figure 2.*

*T-S diagram of November 2012 (left) and October 2015 (right)*

[Figure]

[Referee] Figure 3. This Figure needs improving.
1 The units should be in moles per kg.
2. I am not sure you need to show depths of 500-1000 m since this takes up half the plot and is not discussed much in the text.
3. Please highlight the stations better.
4. I suggest you start other programs to make contour plots in the future (e.g. R) as the ODV palette is not helpful for highlight the data trends that you have here.
*[Response] To be consistent with previous measurements, we will keep the unit, mole per liter for chemical constituents. We will keep the offshore data from 0 – 1000 m because the El Nino effect is thought to occur in the thermocline (upper 1000m), and the discussion (Figure 7 and 8) compared the top 1000 m of water column properties. We increase the contrast of sampling depths and stations.*

[Referee] Figure 4. I cannot see the individual points in the offshore stations so it's hard to determine the extrapolation that has been applied
*[Response] We increase the contrast of sampling depths and stations.*

[Referee] Figure 5. I am not sure this Figure adds value. Figure 5A and Figure5b are very similar to Figure 4a and 4e (identical patterns, just different units. Figure 5c takes up a lot of space and only the bottom right hand section of the chart has any useful data.
*[Response] Figure 5 shows $N_2O$ excess and associated surface fluxes. From there we can see that although surfaces fluxes are low, water column is still oversaturated with N2O, suggesting change of hydrography. Our feeling is that it will be more confusing for the readers not to show Figure 5.*

[Referee] Figure 6. You should connect this figure to the water masses identified in Table 1
*[Response] Good point. Some water masses (oxycline and offshore > 500 m samples) are identified in Table 1. Some clarifying sentences are added in page 9, line 24 – 25, and page 10 line 2.*

[Referee] Figure 8. With no error bars, it is not possible for the reader to know when there is a statistical difference between two depth profiles. On Page 3, Line 13 you say that triplicate measurements were taken, so they should be included.
*[Response] Good point. The analytical precision of $N_2O$ concentration measurement is < 2.5 nmol/L; the precision of previous measurements were assumed to be 5% of the mean concentrations. We added the error bars onto Figure 8 and 9 in the revised manuscript. The precision of our measurements are lower than El Nino variability and we think the conclusion of higher water column $N_2O$ inventories during El Nino will hold.*

[Referee] Table 1. This Table needs improving. 1. Please number the water parcels so they can be easily cross-referenced with the text. 2. Please report depth, O2, and nitrite concentrations for all four identified water parcels. 3. There is no column heading for the third column. 4 What is produced N2O?
*[Response] We will number the water parcels so that readers can cross-reference in the text. We will report depth, oxygen and nitrite concentrations for all water parcels. The heading of third column should be "statistical properties". The term "Produced $N_2O$" is defined in the page 7, line 26 – 27, as "isotopic signature of $N_2O$ produced within the water mass".*

**Anonymous Referee #3**

[Referee] This manuscript presents the distribution and fluxes of N2O in the Eastern Tropical South Pacific region during Oct. 2015, when a strong El Nino event occurred. Measurements of N2O and other related parameters along with its isotopomers were made in the water samples collected from six stations. These measurements have been used to study the variability and biogeochemistry of N2O in the ocean water as well as the effect of this event on the distribution and fluxes of N2O in this region. The manuscript presents important results during this major El Nino event and it is very well written.

However, I have the following clarifications/suggestions for its further improvement. Specific points:

[Referee] 1. The main focus of this manuscript is on the effect of El Nino of the distribution and fluxes of N2O. The three offshore stations show buildup of N2O in the water down to 1000m depth (Fig. 8). However, the comparison for these 3 stations is limited with previous one neutral year only (2012). Also there is large variability in the 0-200m depth. Please show error bars for each point. Measurements for the three coastal stations are compared with the measurements from three different years (2011, 2009 and 1985). All these three stations show very different comparisons. Hence, it is difficult to conclude for the coastal region.

*[Response] The data availability allowed us to compare offshore water column N₂O inventories during between 2015 and 2012. It is the scope of this paper to compare water column properties during El Nino vs. non-El Nino years. In coastal waters, the water column inventories were significantly higher (up to 160%) during El Nino period. These apparent evidence led us to conclude that water column N₂O inventories at lower latitudes during El Nino years were higher than those during non El Nino years (in page 12, line 10). The analytical precision of N₂O concentration measurement is < 2.5 nmol/L; the precision of previous measurements were assumed to be 5% of the mean concentrations. We added the error bars onto Figure 8 and 9 in the revised manuscript. The precision of our measurements are lower than El Nino variability and we think the conclusion of higher water column N₂O inventories during El Nino will hold.*

[Referee] 2. Fig. 9 shows depth integrated N2O concentrations and comparison with earlier measurements. However, the depth taken for each station is limited by earlier measurements and it is different for different stations except for stations B and C. This, in my view, is not correct and gives a wrong comparison. The X axis scale and even the depth for the coastal stations could have been same for all the three stations for a better visualization.

*[Response] We compared the integrated N₂O concentration (N₂O inventories) at the same depth range for each station. The available data allows us to compare depth range down to 1000 m at station A and to 800 m at station B and C. These depth ranges are deeper than El Nino-induced water column changes, generally in the upper 500 meters. For coastal waters, the range is shallower than 250 meters, the entire water column was effected by El Nino. We make clarification in page 11, line 11 – 13.*

[Referee] 3. Are these earlier measurements for the same respective stations? If not, please give their locations also.

*[Response] All the location info for the data presented in Figure 9 are shown in the supplementary material Table S1. Although some measurements were not made at the exact*

*location, data are comparable when measurements were made within 0.75 by 0.75 degree grid. These information are stated in page 5, line 8 – 10.*

[Referee] 4. How the observed decrease in the N2O fluxes compare with earlier studies mentioned in the introduction (P2, L17)?
*[Response] Observation from 2015 – 16 El Nino event showed 23 – 108 µmol/m2/d, 75 – 95 % reduction of fluxes of December 2012 (459 – 1825 µmol/m2/d). We state in page 10, line 23 – 24, this is consistent with observation from Cline et al. (1987) who reported 80% reduction in fluxes.*

[Referee] 5. P1, L25 : 'The depth-integrated N2O....were nearly twice....' is not correct except may be for the E and F stations. Please modify this sentence suitably and also give depth information related to integration.
*[Response] We rewrite part of the sentence as follows: "Water-column inventories of $N_2O$ within the top 1000 m were up to 160% higher than those measured in non-El Niño years,"*

[Referee] 6. How long this El Nino event has been there? The ONI shown in Fig. 1 for 2015 was >0.5 in January itself.
*[Response] Given the definition of El Nino event being ONI > 0.5, the event started in November 2014 and lasted until May 2016. We include this information in page 3, line 5 – 6.*

Minor corrections:
[Referee] P1, L16: '....was developing ..' or developed?
*[Response] The El Nino event was still developing in Oct. 2015, as indicated by ONI in Figure 1*

[Referee] P2, L17: Please change to – '...related to changes in...'
*[Response] Done*

**Anonymous Referee #4**

[Referee] p.1 lines 19-24:  The deepening of the oxycline and deeper $N_2O$ peaks is this just due to a deepening of the isopycnal they occur on, or are they shifting to different density surfaces?
*[Response] The limited data availability does not allow us to conclude either hypothesis. Thus we state our observation in the abstract. It is definitely worth investigating in future cruises. We add this suggestion in page 11, line 30 – 33.*

[Referee] p.3, lines 19-25:  Additional detail on analysis and calibration of N2O isotopologue measurements should be provided.
*[Response] The added information is in page 4, line 13 – 14 and line 20 – 26.*

[Referee] p.4, Equation 5: definition of SP should not have the multiple of 1/2.
*[Response] Indeed. Corrected.*

[Referee] p.6, lines 17-18:  how do you interpret the SP values < 0, especially in the surface which should likely be closer to atmospheric values (∼18‰)?
*[Response] We checked the analysis and we are confident about the measurements. We interpreted as the denitrification dominant $N_2O$ source. Some recent publications showed that $N_2O$ produced by bacterial denitrification has SP values ranging from 0 to minus 10 per mil. Thus we attribute our measurements to denitrification in the water column. We cite these recent results in page 8, line 12.*
*Mothet et al., 2013: http://dx.doi.org/10.1071/EN13021*
*Winther et al., 2018: https://www.biogeosciences.net/15/767/2018/bg-15-767-2018.pdf*

[Referee] p.7, equation 6:  I understand that you are using an end member mixing, or 'Keeling Plot' model for estimating the d value of the produced N2O, but it was less clear how you then attribute the N2O source based on those values.  Please provide additional information, support, justification for the source attributions based on your measurements and calculations.
*[Response] $N_2O$ production pathways can be inferred from SP, because SP is independent of isotopic values of $N_2O$ production substrates; generally, N2O produced via NH4+ oxidation and partial denitrification have distinctive SP values of $30 \pm 5$ ‰ and $0 \pm 5$ ‰, respectively. Simply put, higher produced SP value indicates nitrification whereas lower SP indicates denitrification. It is supposed to be a qualitative, not a quantitative indicator. These above information is stated in page 8, line 1 – 6, and cited Toyoda et al., 2011. The model and calculation is documented by Fuiji et al., 2013 (doi:10.1007/s10872-012-0162-4).*

[Referee] p.7, lines 27-28: it seems difficult to explain a SP value of -9‰ under these conditions. I would be concerned here about calibration, and believe that the authors should further discuss their calibration techniques, and possible explanations for such a low SP value in surface waters.
*[Response] We double checked the isotopic data; our in-house standard (cold seawater) and isotopic references showed no sign of inaccuracies. We are confident that the negative SP values are accurately measured. Some recent publications showed that N2O produced by bacterial denitrification has SP values ranging from 0 to -10 per mil. Thus we attribute our measurements to denitrification in the water column. We cite these recent results in page 8, line 12.*
*Mothet et al., 2013: http://dx.doi.org/10.1071/EN13021*
*Winther et al., 2018: https://www.biogeosciences.net/15/767/2018/bg-15-767-2018.pdf*

[Referee] p.8, line 1: Is there any other evidence of upwelling/diffusion from suboxic zones influencing the upper water column here?

*[Response] Diffusion can occur simply due to a concentration gradient; as the $N_2O$ concentration gradient near the concentration peak can be 3 – 5 nmol/L/m. The study area is also a major upwelling zone. A previous study presented comprehensive results on diffusion and upwelling rate (Haskell et al., 2013). We make clarification in page 8, line 16 – 17.*

[Referee] p. 8, lines 19-22: It is not meaningful to apply the model in equation 6 to an environment in which N2O is being consumed. Is there any evidence of N2O production here, or only consumption?

*[Response] This section requires more clarification. In the oxygen deficient zone where N2O is undersaturated, N2O is being produced and consumed at the same time and reaching a dynamic balance. This is demonstrated by tracer incubation studies (Ji et al., 2015 GRL, doi:10.1002/2015GL066853; Babbin et al., 2015, doi:10.1126/science.aaa8380). The net isotope effects during N2O production and consumption by denitrification is quite significant that both N and O isotopic compositions increased. Therefore it is valid to apply equation 6 to distinguish N2O production. We add clarifying statement in page 8, line 28 – 31.*

[Referee] p.8, line 28: Are you excluding nitrifier-denitrification here, or lumping it in with 'denitrification'?

*[Response] Because nitrifier-denitrification and denitrifier-denitrification have similar SP values, we use 'denitrification' to represent both possible pathways in this sentence. We introduce the above information in page 8, line 4.*

[Referee] p.9, lines 1-3: How would you derive N2O production rates from nitrate and nitrite isotopes?

*[Response] Details can be found in a previous study from Bourbonnais et al., 2017 (doi:10.1002/2016GB005567), in which the authors applied a three-dimensional model to derive N2O production rates from isotopic compositions of N2O, nitrate and nitrite, and fractionation factors during nitrification and denitrification. The above information is added in page 9, line 19 – 22.*

[Referee] p.10: I thought the discussion of El Nino effects on N2O was interesting, and wondered to what extent the sampling period represented conditions during the 2015 El Nino event, and how the 2015 El Nino event might reflect other El Nino events in this region. How broadly applicable are the current results?

*[Response] The cruise in October 2015 was conducted during a developing strong El Nino since early 2015 (see Figure 1). We realized that our study can be treated as a 'snapshot' during El Nino, rather than a continuous record of it. Previous studies (e.g. Stamma et al., 2016, doi:10.5194/os-12-861-2016) have concluded that 2015/16 El Nino can be categorized as "Strong El Nino" based on ONI and other properties. Thus we present comparative study of N2O dynamics between El Nino and non-El Nino years. We clarify this in page 12, line 14.*

[Referee] Figure 1: Why are these years/data in particular the ones compared in this figure? It seems more relevant to compare the strength of the 2015 to other strong El Nino years to gauge how representative were the conditions sampled here of other El Nino periods.

*[Response] Previous studies (e.g. Stamma et al., 2016) have concluded that 2015/16 El Nino can be categorized as "Strong El Nino" based on ONI and other physical properties. Unfortunately, no $N_2O$ concentration or isotope data existed in the database for us to compare the $N_2O$ dynamics between two strong El Nino events. We clarify this in page 5, line 7 – 8.*

[Referee] Figures 3, 4, 5, 7: These section plots are not a particularly effective way to present the data. I understand the desire to present the data in a spatially explicit fashion, but the section plots, with only a few stations, rather large gaps in the data, and considerable smoothing don't help to relay the information to readers. In many of these cases, stacked depth profiles would be a much clearer way of presenting the patterns and comparing to other existing data.

*[Response] The authors have tried a number of ways plotting the data, including stacked depth profiles, to present the water column profiles during 2015/16 El Nino. And the current presentation is rather realistic and effective in conveying information. We also plot delta N2O profiles in 6 separate stations. In some cases there's a large gap between stations, so we leave reasonable blank spaces; we realized the smoothing exerted by the ODV software may be problematic, and we used "weighted-averaged gridding" with contour lines to preserve patterns of depth profiles*

---

## Referee Report (RR1)

Reviewer1 comments

As environmental scientists, our ability to discern temporal (and spatial) variability will be increasingly scrutinized in the very near-future. The authors have improved the manuscript by reporting precision where known. However, I disagree with assuming a precision of 5% from all previous cruises. Nearly all the nitrous oxide datasets derive from Hermann Bange's laboratory and he is a coauthor on this manuscript and I would have thought that some information on precision is available in MEMENTO, but this might not be the case? I am also concerned by the authors response to another Reviewer's comment, copied below. It is unacceptable in other scientific disciplines to simply say 'our values are correct'. For example, transparency of data reporting and validation of datasets through the incorporation of reference material was a major outcome from the GEOTRACES program. The authors should consider attaching their datasets including the calibration, standards, and reference material used to this manuscript. I think in the near future the practice of attaching datasets with the accompanying calibrations and reference material is will become much more common.

**[Referee] p.7, lines 27-28: it seems difficult to explain a SP value of -9‰ under these conditions. I would be concerned here about calibration, and believe that the authors should further discuss their calibration techniques, and possible explanations for such a low SP value in surface waters.**

*[Response] We double checked the isotopic data; our in-house standard (cold seawater) and isotopic references showed no sign of inaccuracies. We are confident that the negative SP values are accurately measured…*

Additional comments

Page 4 Line 7 'The MEMTNTO database did not archive N2O data at the ETSP during previous El Niño events, and therefore we were not able to compare N2O dynamics between two El Niño conditions'.

Spelling of MEMENTO and also doesn't Laura Farias have time-series measurements during El Niño. Are her measurements in MEMENTO or have you reached out to her?

Page 5. Line 27 except that October 2015 had warmer surface water

How much warmer?

Page 10 Line 23 Such a 75 – 95 % reduction in N2O fluxes during the 2015-16 El Niño was consistent with 80% reduction in fluxes observed during 1982-83 El Niño (Cline et al., 1987).

This conflicts with your previous comment. So there is data from El Niño conditions.

---

## Author Response (AR2)

**Response to reviewer**

Note to editor and reviewer: please see the colored texts in the attached manuscript for revisions.

[Referee] As environmental scientists, our ability to discern temporal (and spatial) variability will be increasingly scrutinized in the very near-future. The authors have improved the manuscript by reporting precision where known. However, I disagree with assuming a precision of 5% from all previous cruises. Nearly all the nitrous oxide datasets derive from Hermann Bange's laboratory and he is a coauthor on this manuscript and I would have thought that some information on precision is available in MEMENTO, but this might not be the case?

[Response] Per reviewer's request, we retrieved the analytical precision of $N_2O$ concentration measurements from previous publications (Friederich et al., 1985; Baustian et al., 2012; Kock et al., 2016), and revised the error bars on figure 8 and 9. These analytical precisions were derived from the standard deviations of triplicate measurements and are generally better than 5% of mean concentration values. We will include these precision values in the supplementary dataset. It should be noted, however, that the MEMENTO database does not include measurement error at the moment. The sentences on page 5, line 10 – 12 are rewritten as: "Standard deviation of repeated $N_2O$ concentration measurements (analytical precision) for archived $N_2O$ concentration datasets were retrieved from respective references (see supplementary Table S1). These analytical precisions are < 5% of $N_2O$ concentration values."

[Referee] I am also concerned by the authors response to another Reviewer's comment, copied below. It is unacceptable in other scientific disciplines to simply say 'our values are correct'. For example, transparency of data reporting and validation of datasets through the incorporation of reference material was a major outcome from the GEOTRACES program. The authors should consider attaching their datasets including the calibration, standards, and reference material used to this manuscript. I think in the near future the practice of attaching datasets with the accompanying calibrations and reference material is will become much more common.

**[Referee] p.7, lines 27-28: it seems difficult to explain a SP value of -9‰ under these conditions. I would be concerned here about calibration, and believe that the authors should further discuss their calibration techniques, and possible explanations for such a low SP value in surface waters.**

*[Response] We double checked the isotopic data; our in-house standard (cold seawater) and isotopic references showed no sign of inaccuracies. We are confident that the negative SP values are accurately measured…*

[Response] There were four isotopic/isotopomeric reference gases used in the measurements. Their respective isotopic/isotopomeric values are as follows:

| Standard name | Calibrated isotopic value | | | | |
|---|---|---|---|---|---|
| | $\delta^{15}N_\alpha$ | $\delta^{15}N_\beta$ | $\delta^{15}N_{bulk}$ | SP | $\delta^{18}O$ |
| CB09715 | -82.14 | -78.02 | -80.08 | -4.12 | 21.64 |
| CB09766 | 5.55 | -12.87 | -3.66 | 18.42 | 32.73 |
| CB08976 | 15.7 | -3.21 | 6.25 | 18.91 | 35.16 |
| 53504 | 1.71 | 94.44 | 48.08 | -92.73 | 36.01 |

The above table will be incorporated into the supplementary material. The reference materials encompass the isotopic/isotopomeric values reported in this manuscript, especially the low SP values (-10 – 0 ‰) from the coastal waters. The calibration was thus straightforward and followed standard practices. There is no reason to believe the reported low SP values were due to calibration error. On page 8, line 6 – 7 and line 13 – 14, we explained that these low SP values are indicative of strong denitrifying activities, which were demonstrated recently in culture (Winther et al., 2018) and river water (Mothet et al., 2013).  Per the reviewer's request, we are attaching the N2O isotopic/isotopomeric dataset reported in this manuscript.

Additional comments

[Referee] Page 4 Line 7 'The MEMTNTO database did not archive N2O data at the ETSP during previous El Niño events, and therefore we were not able to compare N2O dynamics between two El Niño conditions'.
Spelling of MEMENTO and also doesn't Laura Farias have time-series measurements during El Niño. Are her measurements in MEMENTO or have you reached out to her?

[Response] The spelling is corrected on page 5, line 7. We stated that "The MEMENTO database has not archived any $N_2O$ datasets in the ETSP region during previous El Niño events," on page 5, line 7 – 8.  The N2O dataset presented by Farías et al. 2015 (doi:10.1088/1748-9326/10/4/044017) was from one time series station at 36.5 °S, 73.13 °W, outside of ETSP region. And the time frame of that dataset did not cover an El Niño period (2002 – 2012). Unless the reviewer provides a doi for an article or a dataset, we cannot think of an N2O dataset in the ETSP during previous El Niño.

[Referee] Page 5. Line 27 except that October 2015 had warmer surface water
How much warmer?

[Response] We added 2 – 4 °C on page 5, line 28.

[Referee] Page 10 Line 23 Such a 75 – 95 % reduction in N2O fluxes during the 2015-16 El Niño was consistent with 80% reduction in fluxes observed during 1982-83 El Niño (Cline et al., 1987). This conflicts with your previous comment. So there is data from El Niño conditions.

[Response] The N2O dataset presented by Cline et al. 1987 was from central equatorial Pacific and has no overlap with the ETSP region. However their finding of ~80% reduction in fluxes was comparable to our estimate (75 – 95%) in the ETSP. The sentence on page 10, line 26 – 28 was not clear; and we revised the sentence as "Such a 75 – 95 % reduction in $N_2O$ fluxes during the 2015-16 El Niño in the ETSP was comparable to an 80% reduction in fluxes observed in the central equatorial Pacific during the 1982-83 El Niño (Cline et al., 1987). "

In addition, we have made the following revisions to improve the clarity of the manuscript:

Page 1, line 16, add "in the water column".

Page 1, line 26, add "At multiple stations,".

Page 2, line 27 – 29. The sentence is rewritten as: "Finally, the effects of a strong El Niño event on the surface and water column $N_2O$ distributions were investigated by incorporating archived ETSP datasets that demonstrated contrasting hydrography and biogeochemistry between El Niño and non- El Niño years.".

Page 8, line 3, add "is only determined by".

Page 12, line 15, add "near the suboxic waters".

Page 12, line 17, replace with "a decrease of water column oxygen consumption".

[revised manuscript text omitted]